# Spatiotemporal patterns and drivers of wildfire $CO_2$ emissions in China from 2001 to 2022

Xuehong Gong[1,2], Zeyu Liu[2,3], Jie Tian[2,3], Qiyuan Wang[2,3], Guohui Li[2,3], Zhisheng An[1,2], Yongming Han[2,3*]

[1] State Key Laboratory of Earth Surface Processes and Resource Ecology, Faculty of Geographical Science, Beijing Normal University, Beijing 100875, China
[2] State Key Laboratory of Loess and Quaternary Geology, Institute of Earth Environment, Chinese Academy of Sciences, Xi'an 710061, China
[3] National Observation and Research Station of Regional Ecological Environment Change and Comprehensive Management in the Guanzhong Plain, Xi'an 710061, China

*Correspondence to*: Yongming Han (yongming@ieecas.cn)

**Abstract.** Wildfires release large amounts of greenhouse gases into the atmosphere, exacerbating climate change and causing severe impacts on air quality and human health. In this study, based on a bottom-up approach and using satellite data, combined with emission factor and aboveground biomass data for different vegetation cover types (forest, shrub, grassland, cropland), the dynamic changes in $CO_2$ emissions from wildfires in China from 2001 to 2022 were analyzed. The results showed that between 2001 and 2022, the total $CO_2$ emissions from wildfires in China were 937.7 Tg (522.6-1516.0 Tg, 1 Tg = $10^{12}$ g), with an annual average of 42.6 Tg (23.8-68.9 Tg). The $CO_2$ emissions from cropland and forest fires were relatively high, accounting for 45% and 46% of the total, respectively. The yearly variation in $CO_2$ emissions from forest and shrub fires showed a significant downward trend, while emissions from grassland fires remained relatively stable. In contrast, the $CO_2$ emissions from cropland fires showed an upward trend, primarily in Northeast China. Hotspot analysis and Geographically and Temporally Weighted Regression (GTWR) models revealed significant spatial heterogeneity in emissions across vegetation types. Persistent hotspots of shrub and forest fires were located in Southwest and South China, while Northeast China experienced sporadic but extreme fire events. The GTWR model for shrub fire $CO_2$ emissions exhibited the highest predictive performance ($R^2 = 0.87$) and climatic factors (particularly temperature and humidity) were the main influencing factors. Notably, the recent rise in cropland fire $CO_2$ emissions in Northeast China is closely linked to region-specific straw burning policies. The research results provide valuable references for atmospheric transport models, regional fire management, and national carbon accounting frameworks in the context of climate change.

## 1 Introduction

To limit the global average surface temperature rise to 1.5 °C higher than preindustrial levels, carbon dioxide ($CO_2$) emissions must reach net zero by mid-century through various pathways (Rogelj et al., 2018). Globally, wildfires reduced carbon storage in vegetation by approximately 10% from 2001 to 2012 (Lasslop et al., 2020). This significantly impacted the

concentration of $CO_2$ in the atmosphere (Langenfelds et al., 2002; van der Werf et al., 2004; Wotawa and Trainer, 2000). According to Global Wildfire Information System data compiled by Our World in Data (Our World in Data, 2025), global wildfire $CO_2$ emissions have increased since 2020, fluctuating between 5 and 7 Gt $CO_2$ per year (1 Gt = $10^{15}$ g), with record-high levels observed in 2021 and 2023. However, the role of wildfires as a critical factor in carbon sinks and sources is often overlooked. To mitigate climate change and fully understand the carbon exchange mech.anisms between terrestrial ecosystems and the atmosphere, it is essential to consider the impa0.c0ts of wildfire $CO_2$ emissions on the Earth system (Chuvieco et al., 2019; Giglio et al., 2018; Kasischke et al., 1995; McGuire et al., 2001; Zhang et al., 2013).

The significant differences in global wildfire $CO_2$ emissions among countries highlight the complexity of wildfire $CO_2$ emissions. Extreme forest fires in several countries, such as Australia, Canada, and the United States, often release $CO_2$ that exceeds the cumulative $CO_2$ emissions of several years in the same region, significantly impacting the global climate and the environment. Boreal fires, which usually contribute 10% of global fire $CO_2$ emissions, accounted for 23% in 2021 (0.48 Gt C), marking the highest fraction since 2000 (Zhang et al., 2023). The unprecedented wildfires in Canada in 2023 released significant amounts of air pollutants and greenhouse gases into the atmosphere. Simulation results (Wang et al., 2023) have indicated that these wildfires emitted more than 1.3 Gt $CO_2$ and 0.14 Gt $CO_2$ equivalent of other greenhouse gases, including $CH_4$ and $N_2O$. The greenhouse gas emissions associated with wildfires exceeded twice the planned cumulative anthropogenic emissions reductions in Canada over a decade. Shiraishi et al. (2021) used a bottom-up approach to estimate $CO_2$ emissions from catastrophic fires in Australia between 2019 and 2020. The results showed that from March 2019 to February 2020, Australia's annual $CO_2$ emissions were estimated to be $806 \pm 69.7$ Tg (1 Tg = $10^{12}$ g) $CO_2$ year$^{-1}$, equivalent to 1.5 times its total greenhouse gas emissions ($CO_2$ equivalent) in 2017. Phillips et al. (2022) reported that by the middle of this century, wildfires in northern North America could lead to a cumulative net source of approximately 12 Gt $CO_2$, accounting for approximately 3% of the remaining global $CO_2$ emissions, which is closely related to the temperature targets of the Paris Agreement. In the context of climate change, wildfires are becoming more frequent, and $CO_2$ emissions from wildfires are often influenced by human intervention. Phillips et al. (2022) found that increasing investment in fire management to avoid $CO_2$ emissions is equivalent to or lower than other mitigation strategies. Therefore, changes in fire management may impact global atmospheric $CO_2$ concentrations, and proactive management strategies effectively reduce $CO_2$ emissions (Kelly et al., 2013; Phillips et al., 2022; Van Wees et al., 2021). Despite the growing importance of wildfire $CO_2$ emissions in climate change, such emissions are often excluded from international climate frameworks, including national inventories under the United Nations Framework Convention on Climate Change (UNFCCC), due to their classification as "natural disturbances" in the Intergovernmental Panel on Climate Change (IPCC) guidelines for Land-Use Change and Forestry (LULUCF) (IPCC, 2019).

China has released numerous wildfire emission inventory, but previous research on wildfire emissions in China has focused chiefly on small-scale regions and short-term periods (Cao et al., 2005; Huang et al., 2012; Qiu et al., 2016; Tian et al., 2011; Wu et al., 2018). Wang and Zhao (2008) established an atmospheric pollutant emission inventory of cropland fires in China in 2006 using the emission factor method and analyzed its spatiotemporal distribution characteristics. Wu et al. (2018)

estimated pollutant emission inventories from wildfires in central and eastern China from 2003 to 2015 using remote sensing images but did not include the heavily polluted northeast region. In addition, most studies have focused mainly on atmospheric pollutant emissions, with limited research on $CO_2$ emissions (Jin et al., 2022; Wang and Zhao, 2008; Xie et al., 2024; Yin et al., 2019). Xie et al. (2024) used the GEOS-Chem model to investigate the impact of cropland fires on severe haze events in

Heilongjiang Province. They reported high uncertainty in the existing Global Fire Emissions Database (GFED) version 4.1 emission inventory. Van der Werf et al. (2017) also noted substantial uncertainty in estimating wildfire emissions in existing emission inventories. Consequently, there is a critical need to quantify the long-term dynamics of wildfire $CO_2$ emissions across diverse vegetation types.

Traditionally, wildfire emission inventories using population or cropland area weights to allocate total emissions to grid

cells have high uncertainties (Streets et al., 2003; Zhang et al., 2013). With the advancement of remote sensing technology, recent studies have shifted to satellite-based estimation methods, using active fire detection or burned area datasets to improve spatial accuracy. Inventories such as GFED (Chen et al., 2023) and the NCAR Fire Inventory (FINN) (Wiedinmyer et al., 2011) rely on satellite-derived fire count data (e.g., active fire product MCD14 ML) or burned area products (e.g., MCD64A1) to infer the timing and location of fire emissions (Giglio et al., 2016, 2018). Although satellite remote sensing has greatly

improved the spatial and temporal resolution of fire detection, several practical challenges remain. For example, cloud cover, satellite overpass intervals, fire intensity thresholds, and pixel resolution can result in the underdetection of short-duration or low-intensity fires. To mitigate these limitations, this study integrated multi-source satellite products to enhance the completeness of the fire signal. Additionally, many existing global inventories rely on globally aggregated vegetation data (such as global land cover, and biomass), which further introduces errors, especially in transition zones between cropland and

natural vegetation (e.g., forest-agricultural mosaics), where misclassification may lead to overestimation or underestimation of fire emissions.

To overcome these shortcomings, this study integrated China's regionally validated vegetation cover datasets (Xu et al., 2018), multi-source burned area satellite products, and regionally derived biomass data (Hu et al., 2006; Su et al., 2016; Yin et al., 2023) to develop a 500-meter resolution wildfire $CO_2$ emission inventory for China (2001-2022). Additionally, we used

spatially weighted regression models to explore the drivers of emission variability and analyzed the impacts of national fire management policies on $CO_2$ emissions. The findings provide insights into the role of governance in shaping fire emissions and offer useful references for future wildfire management strategies. This multi-year emission inventory can also be used in atmospheric transport models to support the development of effective global warming mitigation strategies.

## 2 Data and methods

### 2.1 Study area

China is located in the eastern part of the Eurasian continent on the west coast of the Pacific Ocean. It spans approximately 50 degrees of latitude (3-53 °N) from north to south and 60 degrees of longitude (73-135 °E) from east to west, with a land

area of approximately $9.60 \times 10^6$ km². There are differences in the geographical distribution of cropland, grassland, shrubs, and forests in China. In this study, China was divided into seven subregions based on geographic and ecological characteristics: Northeast China (NE), North China (NC), Central-West China (CW), South China (SC), Southwest China (SW), Northwest China (NW), and the Tibetan Plateau (TP) (Fig. 1). Croplands are mainly located in the eastern plains and coastal areas, especially in NE ( provinces such as Heilongjiang, Jilin, and Liaoning), and NC (provinces such as Hebei, Henan, Shandong, and Jiangsu), where the terrain is flat and suitable for agriculture. Grasslands are mainly distributed across the Inner Mongolia region (spanning NE and CW), the Xinjiang region of NW, and parts of SW. Forests and shrubs are primarily concentrated in NE (especially Heilongjiang), SW (provinces such as Yunnan, Guizhou, and Sichuan), and SC (provinces like Jiangxi and Hunan).

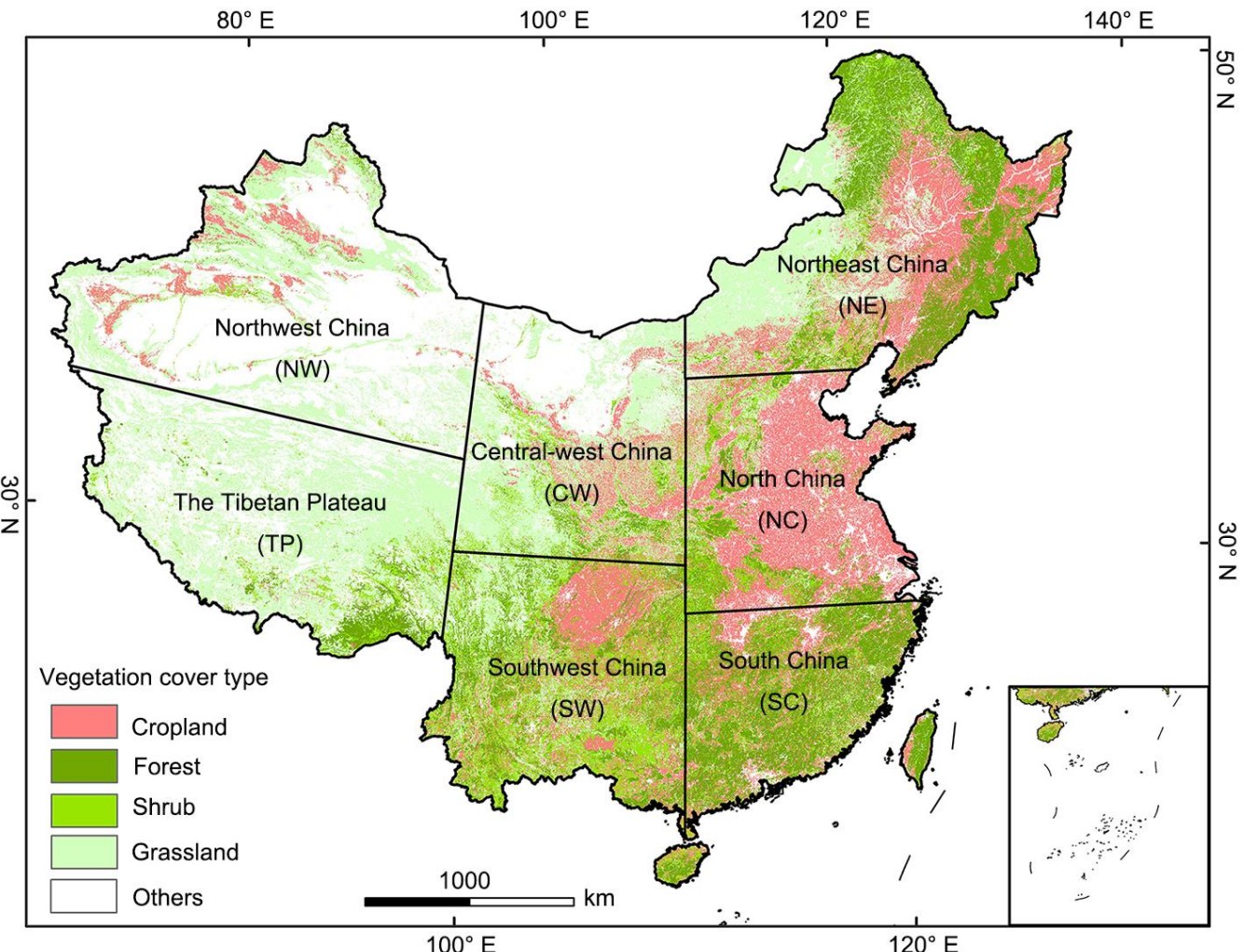

**Figure 1: Regional divisions and vegetation distribution in China. The seven regions include: North China (NC): 109°E to eastern border, 30-41°N; South China (SC): 109°E to eastern border, south border to 30°N; Southwest China (SW): 100-109°E, south border to 32°N; Central-West China (CW): 100–109°E, 32°N to northern border; Northeast China (NE): 109°E to eastern border, 41°N to**

**northern border; Northwest China (NW): Western border to 100°E, 36°N to northern border; the Tibetan Plateau (TP): Western border to 100°E, south border to 36°N.**

## 2.2 CO₂ emission estimation

In this study, we employed a bottom-up approach to develop an inventory of China's wildfire $CO_2$ emissions. Wildfire $CO_2$ emissions were calculated using the following formula:

$$E_i = \sum_{x,i} BA_{x,i} \cdot\times\cdot AGB_{x,i} \cdot\times\cdot CE_i \cdot\times\cdot EF_i \tag{1}$$

where the subscripts $x$ and $i$ represent the grid cell and vegetation cover type (forest, shrub, grassland and cropland), respectively. The vegetation cover data were sourced from the China Land Use Land Cover Remote Sensing Monitoring Dataset (CNLUCC) (Xu et al., 2018). and a 1 km harvesting area dataset for three staple crops (e.g., corn, wheat, and rice) in China from 2000 to 2019 was obtained from Luo et al. (2020). $E_i$ represents the $CO_2$ emissions, $BA_{x,i}$ is the burned areas (ha), $AGB_{x,i}$ is the aboveground biomass (t ha⁻¹), and $EF_i$ is the emission factor. $CE_i$ is the combustion efficiency. All datasets were resampled to 500 m resolution to ensure spatial consistency.

### 2.2.1 Burned areas

BA for each vegetation cover type was primarily estimated using the MODIS-MCD64A1 product (Giglio et al., 2018), which provides global monthly burned area estimates. However, it is well acknowledged that MODIS-MCD64A1 tended to underestimate small and fragmented fires. To address this issue, we applied scaling factors ($\alpha_i$) to correct the MODIS-derived BA estimates. The scaling factors were derived from the comparison of MODIS-derived BA with two independent global burned area datasets: the FireCCI51 product (Lizundia-Loiola et al., 2020) released by ESA (http://cci.esa.int/data) and a global GFED500 product (Van Wees et al., 2022), with their specific values provided in Table S1. The corrected burned area for each vegetation cover type was obtained by multiplying the MODIS-derived BA values by the corresponding scaling factor. This correction accounts for the known systematic underestimation of small and fragmented fires by the MODIS MCD64A1 product.

$$BA_{\text{corrected},i} = BA_{\text{MODIS},i} \times \alpha_i \tag{2}$$

where $i$ denotes vegetation type.

### 2.2.2 Emission factors

The emission factor refers to the gas released per unit mass of dry combustible material during combustion, typically in grams per kilogram (g kg⁻¹). This is a crucial parameter for calculating gas emissions during biomass burning, such as $CO_2$, methane ($CH_4$), and carbon monoxide (CO). Emission factors are influenced by various factors, including the combustibility of tree species, differences in vegetation cover types, and the intensity of flame combustion (Andreae and Merlet, 2001; Lü et al., 2006). To ensure the accuracy of the wildfire emission inventory as much as possible, it is essential to choose appropriate emission factors. This study comprehensively analyzed many studies in the literature to summarized the emission factors of

$CO_2$ generated by wildfires under different vegetation cover types, as listed in Table S2. Finally, the average values from the literature were selected as the emission factors of the different vegetation cover types.

### 2.2.3 Aboveground biomass

Previous studies have mainly used the aboveground biomass data from Fang et al. (1996) for forests. Forest aboveground biomass data in recent years need to be updated. In this study, the aboveground biomass data of forests from 2001 to 2012 were obtained from Su et al. (2016). The data for 2013 to 2022 were obtained from Yan et al. (2023). For shrub, Chinese local biomass density data were collected in Table S3 (Hu et al., 2006). Grassland aboveground biomass was calculated using the exponential model by Gao et al. (2012):

$$AGB_{grass} = 20.1921 \times e^{3.2154 \times (NDVI)} \tag{3}$$

where $AGB_{grass}$ is the aboveground biomass of grassland (g m$^{-2}$) based on the average normalized difference vegetation index (NDVI) value of the growing season. NDVI data were sourced from China's regional 250 m normalized difference vegetation index dataset (Gao et al., 2024b).

To determine the aboveground biomass of cropland, we gathered the crop-specific yield per unit area of different crops from the China Statistical Yearbook (NBSC, 2001-2022). The aboveground biomass burned in the field of cropland from major crops is calculated from the crop-specific yield per unit area, the straw production rates and the dry matter content of each crop residue as follows:

$$AGB_i = P_i \times R_i \times D_i \tag{4}$$

where $i$ represented crop type (rice, corn, wheat and other crops), $AGB_i$ was the aboveground biomass of cropland burned in the field (g m$^{-2}$); $P_i$ was the crop-specific yield per unit area (g m$^{-2}$), $R_i$ was the straw yield ratio for each crop type; $D_i$ is the dry matter content of each crop residue. The other crops were defined as the average of rice, corn, and wheat. For each crop type, data for R and D were collected from published literature (Table S4).

### 2.2.4 Combustion efficiency

The combustion efficiency (CE) of biomass is a crucial factor determining the accuracy of wildfire $CO_2$ emissions estimates. It is influenced by multiple factors, including fire intensity, wildfire type, moisture content and load of combustibles, as well as meteorological conditions. Hély et al. (2003) established an empirical relationship between combustion efficiency and vegetation cover fraction (FVC), which was applied in this study to calculate the CE for forests and grasslands. The FVC used in this study was sourced from China's regional 250 m fractional vegetation cover dataset (Gao et al., 2024a). For regions with an FVC exceeding 60%, the CE values for forest and grassland were set at 0.3 and 0.9, respectively. When the FVC was below 40%, the CE values for forest and grassland were 0 and 0.98, respectively. In areas where the vegetation cover ranged from 40% to 60%, the CE for forest cover was defined as 0.3. The CE for grassland was calculated using the following formula:

$$CE = e^{-0.13 \times FVC} \tag{5}$$

The CE of shrub was set at 0.7, based on a China-specific literature and global biomass burning studies (Junpen et al., 2020; Mieville et al., 2010; Ping et al., 2021; Van Leeuwen et al., 2014; Zhou et al., 2017). The CE of corn, wheat, and rice was obtained from He et al. (2015), with values of 0.92, 0.92, and 0.93, respectively. The CE of other crops was taken as the average value for corn, wheat, and rice (i.e., 0.923).

It is important to note that although the CE values for different vegetation types were carefully selected based on comprehensive literature reviews, CE is inherently variable and can differ significantly across various combustion phases. Since this study aimed to estimate emissions over extended periods (ranging from months to years), the adopted CE values represent average combustion conditions rather than instantaneous ones. This averaging approach may introduce uncertainties in the emission estimates, especially in scenarios where rapid changes in combustion efficiency occur.

## 2.3 Spatiotemporal analysis of wildfire $CO_2$ emissions

### 2.3.1 Global spatial autocorrelation analysis

Global spatial autocorrelation is a fundamental concept in spatial statistics, used to assess the overall spatial dependence of a variable across a study region. Anselin's Moran's I index (Anselin, 1995; Moran, 1948) and the Getis-Ord Gi coefficient* (Getis and Ord, 1992) are commonly used to measure the degree of spatial clustering and heterogeneity. Moran's I is a global spatial autocorrelation statistic that quantifies the degree to which similar attribute values are clustered or dispersed in space. The Moran's I is calculated as follows:

$$I = \frac{n}{S_0} \times \frac{\sum_{i=1}^{n}\sum_{j=1}^{n} w_{ij}(x_i-\bar{x})(x_j-\bar{x})}{\sum_{i=1}^{n}(x_i-\bar{x})^2} \tag{6}$$

where I is the Global Moran's I index, n is the total number of spatial elements; $x_i$ and $x_j$ are the observed values at spatial units $i$ and $j$, respectively; $\bar{x}$ is the mean of all observed values; $w_{ij}$ is the weight matrix for the adjacency relationships between geographical units; $S_0$ is the sum of all spatial weights.

The Moran's I is between -1 and 1. A value of I > 0 indicates positive spatial autocorrelation, i.e., similar values (high or low) tend to occur near each other, while I < 0 indicates dissimilar values are adjacent. I ≈ 0 suggests a random spatial pattern. Statistical significance is assessed by comparing the observed Moran's I to a null distribution generated via random permutations. A $z$-score > 2.58 and $p$-value < 0.01 indicates a statistically significant spatial clustering pattern at the 99% confidence level. In the context of this study, significantly positive Moran's I values indicate that wildfire $CO_2$ emissions are spatially clustered, meaning that regions with high emissions tend to be adjacent to other high-emission areas, and low-emission regions are likewise grouped. This justifies further localized analyses such as hotspot detection.

### 2.3.2 Hot spot analysis

While Moran's I provides a global measure of spatial autocorrelation, it does not explicitly identify localized clusters of high or low values. To address this limitation, the Getis-Ord Gi* statistic (Getis and Ord, 1992) is commonly used to identify

statistically significant hotspots and coldspots within spatial datasets. Unlike Moran's I, which captures both positive and negative spatial autocorrelation, the Gi* statistic focuses on detecting concentration patterns of high or low values within the study area. The Getis-Ord Gi* statistic is defined as:

$$G_i^* = \frac{\sum_j w_{ij} x_j - \bar{x} \sum_j w_{ij}}{S \sqrt{\left[\frac{n \sum_j w_{ij}^2 - \left(\sum_j w_{ij}\right)^2}{n-1}\right]}}$$
(7)

Where $G_i^*$ is the Getis-Ord Gi* statistic for location $i$; $x_j$ is the observed value at location $j$ (e.g., $CO_2$ emissions); $\bar{x}$ is the global mean of the observed variable; $w_{ij}$ is the spatial weight matrix, representing the spatial relationship between locations $i$ and $j$; $n$ is the total number of spatial units; $S$ is the standard deviation of the observed values.

The Gi* statistic is essentially a ratio that compares the local sum of a variable within a specified distance to the global sum, adjusted for the number of spatial units and their spatial relationships. High positive Gi* values indicate clusters of high values (hotspots), while low negative Gi* values indicate clusters of low values (coldspots). Locations with Gi* values near zero indicate random spatial patterns without significant clustering. Statistical significance is assessed using Z-scores and corresponding $p$-values. In this study, Gi* analysis was used to detect persistent high- and low-emission clusters of wildfire $CO_2$ emissions across China from 2001 to 2022. The results provided spatially explicit insights into emission patterns.

### 2.3.3 Geographically and temporally weighted regression model

To capture the spatial and temporal variations of the drivers of wildfire $CO_2$ emissions, the Geographically and Temporally Weighted Regression (GTWR) model was used (Huang et al., 2010). Unlike traditional global regression models, GTWR allows the coefficients of explanatory variables to vary across both space and time, providing a more precise estimation of the local influence of different driving factors. The GTWR model is defined as:

$$y_i = \beta_0(u_i, v_i, t_i) + \sum_k \beta_k(u_i, v_i, t_i) x_{ik} + \epsilon_i$$
(8)

Where $x_i$ is the response variable (wildfire $CO_2$ emissions); $(u_i, v_i, t_i)$ are the spatial coordinates and timestamp for location $i$; $\beta_0$ is the intercept term; $\beta_k$ is the local coefficient for the $k$th explanatory variable; $x_{ik}$ is the $k$th explanatory variable; $\epsilon_i$ is the error term. The accuracy of the GTWR model depends significantly on the choice of bandwidth and kernel function, which control the spatial and temporal influence of neighboring observations. In this study, an adaptive bandwidth was used to ensure that each observation has a sufficient number of neighbors, while a tricube kernel was selected for its smooth distance decay function. The optimal bandwidth was determined using the corrected Akaike Information Criterion (AICc), a widely used criterion for model selection that balances model complexity and goodness of fit (Hurvich et al., 1998; Hurvich and Tsai, 1989). This approach enabled us to explore how the effects of climatic and socioeconomic variables on wildfire emissions vary across regions and over time.

# 3 Results and discussion

## 3.1 Interannual variation in CO₂ emissions

The total CO₂ emissions from wildfires in China from 2001 to 2022 were 937.7 (522.6-1516.0 Tg) Tg, with an average annual value of 42.6 (23.8-68.9) Tg, CO₂ emissions from wildfires in China were relatively low, decreasing slowly by 0.6 Tg per year (Fig. 2a). CO₂ emissions from cropland and forest fires were relatively high, accounting for 45% and 46% of the total wildfire emissions in China, respectively; shrub fires emissions account for 8% of the total wildfire emissions in China, while grassland fire emissions were the lowest, accounting for only 2% of the total wildfire emissions in China (Fig. 2b).

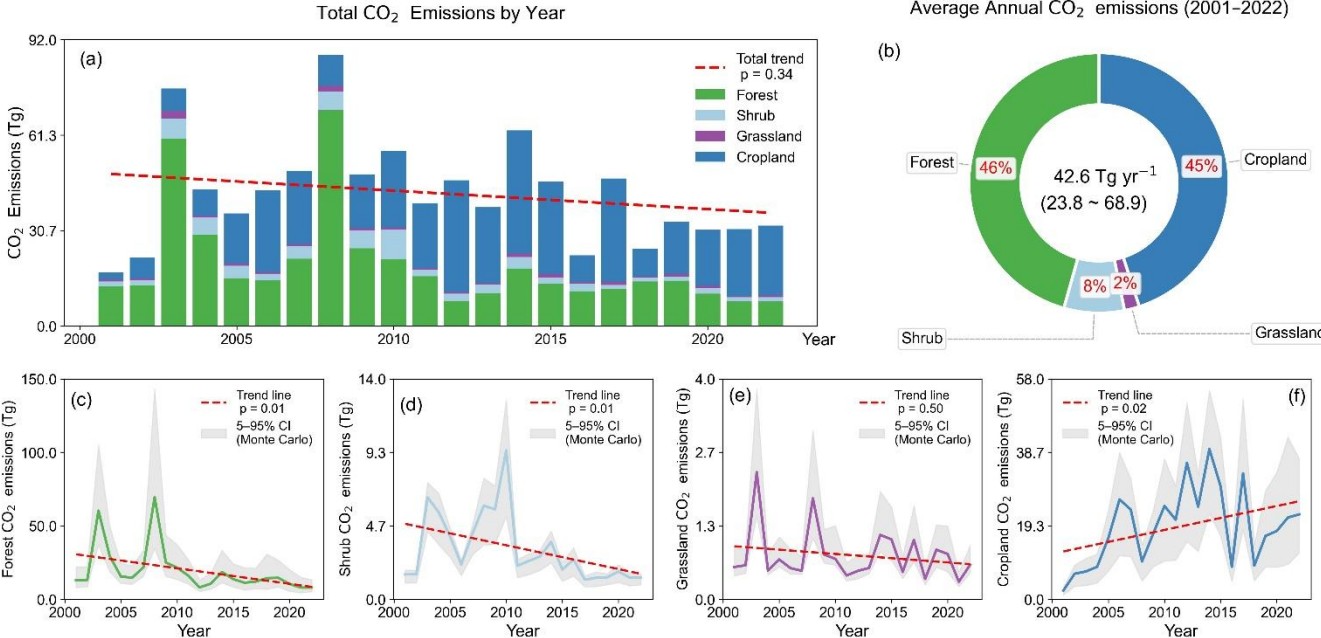

**Figure 2: (a) Annual CO₂ emissions within specific vegetation cover types from 2001 to 2022 in China; (b) Contribution of different vegetation cover types to the total CO₂ emissions from 2001 to 2022 in China. (c-f) Time series of CO₂ emissions for forest, shrub, grassland and cropland, respectively; the red dashed line is the linear trend and the grey shaded envelope represents the 5th-95th percentile confidence interval from Monte Carlo uncertainty analysis; the *p*-values are derived from the Mann-Kendall trend test, a non-parametric statistical method used to assess the presence of a monotonic (increasing or decreasing) trend in a time series without assuming any specific data distribution. A *p*-value < 0.05 indicates a statistically significant trend at the 95% confidence level.**

The annual CO₂ emissions from different types of fires showed varying temporal trends. The downward trend for forest and shrub emissions was significant, with a decrease of 1.1 and 0.2 Tg per year, respectively (Fig. 2c and 2d). Such a decline may reflect effective forestry management strategies for forest and shrub fires (Fig. 12). In contrast, cropland emissions showed a upward trend, with an annual increase of 0.6 Tg (Fig. 2f). This may be attributed to the increased agricultural intensity and straw production in major grain-producing regions, particularly in northeastern China. Additionally, shifts in local open-field burning management strategies, such as the introduction of temporally concentrated burning windows, may have enhanced the

detectability of agricultural fires via remote sensing. The emission trend for grassland was relatively stable (Fig. 2e), which might be influenced by a combination of ecological and anthropogenic factors.

## 3.2 Monthly variation in $CO_2$ emissions

The $CO_2$ emissions from different vegetation cover types showed significant seasonal fluctuations, with certain months
showing higher emissions than others (Fig.3a). Wildfires had lower $CO_2$ emissions in July and August, which may correspond to the respective wet seasons (Fig. 3a). Forest, shrub, and grassland fire $CO_2$ emissions had higher emissions in February, March, and April, possibly related to the dry weather and accumulation of combustible materials in spring, increasing the risk of fires (Fig. 3b-d). The extreme fire $CO_2$ emissions observed in 2003 and 2008 were both associated with prolonged drought conditions during the spring season. Cropland fire $CO_2$ emissions showed significant emission peaks in April, May, and June
(Fig. 3e). This pattern may be related to specific agricultural activities (such as plowing, sowing, and harvesting) cycles, as cropland fires often occur after harvest when crop residues are burned to prepare for the next planting season. The spatial distribution of forest, shrub, and grassland fire emissions was relatively similar among the different months (Figs. S1-S3). In contrast, the spatial distribution of $CO_2$ emissions from cropland fires varied significantly across different months and was likely influenced by policy management (Fig.4). High emissions in March and April were concentrated in NE region, while
emissions in May and June were primarily associated with the NC region. The regional difference in peak emission months can be attributed to distinct cropping systems and climatic conditions. In the NE region (e.g., Heilongjiang and Jilin), cold winters and delayed spring thaw often push straw burning activities into March-April, following the autumn harvest. In contrast, the NC region (e.g., Anhui, Henan, Jiangsu) practices a double-cropping system of winter wheat and summer maize, where wheat is harvested in May-June, and burning of straw residues is typically observed during this transition period.

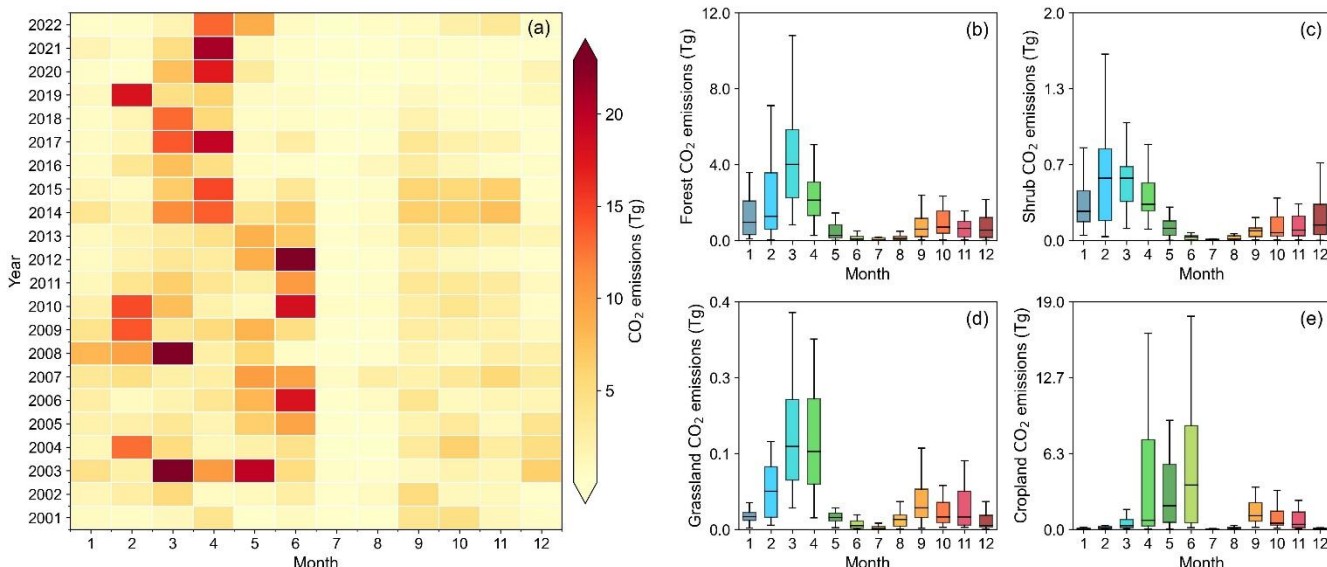

**Figure 3: (a) Heatmap of monthly total wildfires $CO_2$ emissions in China from 2001 to 2022; box plots of $CO_2$ emissions for specific vegetation cover types per month from 2001 to 2022 in China, showing the median (black line), mean (box), and the range within 1.5 times the interquartile range (IQR): (b) Forest, (c) Shrub, (d) Grassland, (e) Cropland.**

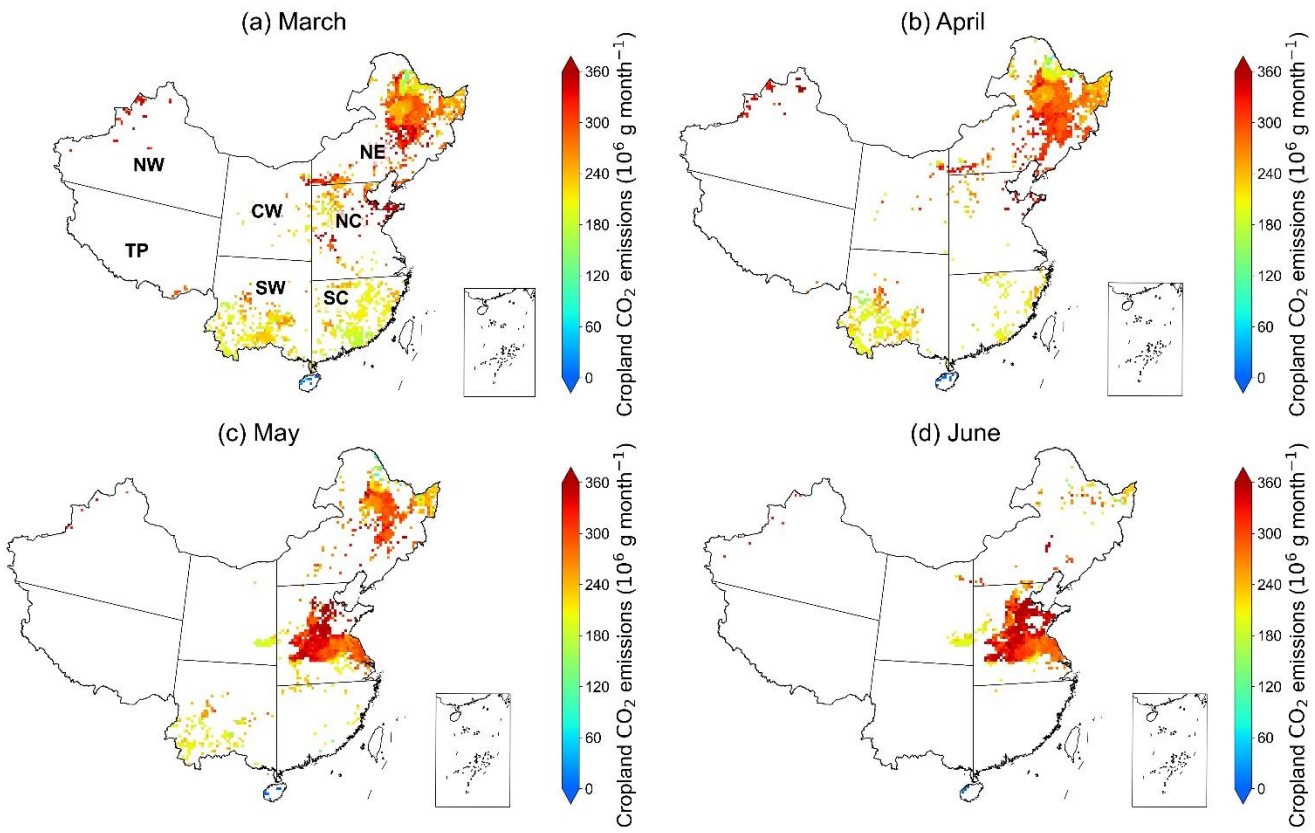

**Figure 4: Spatial distribution of monthly $CO_2$ emissions within cropland fires from 2001 to 2022 in China: (a) March, (b) April, (c) May, and (d) June. The original emission data at 500 m resolution were resampled to 0.25° grid to enhance visual clarity.**

### 3.3 Spatiotemporal variations in $CO_2$ emissions

Due to differences in geographical location, climate conditions, and population density, the spatiotemporal distribution of $CO_2$ emissions in each region exhibits heterogeneity (Fig. 5). emissions in NW, CW, and TP regions were relatively low, accounting for only 3% of China's annual average emissions from 2001 to 2022. In contrast, regions with high emissions were mainly concentrated in NE, NC, SC, and SW regions (Fig. 5a).

To assess whether wildfire $CO_2$ emissions exhibit statistically significant spatial clustering patterns at the national scale, we first applied Moran's I. This step was crucial because it determined the necessity of subsequent local cluster analyses (such as hotspot analysis). The results (Table 1) showed significantly positive Moran's I values for fire emissions across all vegetation types ($I > 0, p < 0.01; Z > 2.58$), indicating non-random spatial distributions and strong global spatial autocorrelation. These findings supported the use of the Getis-Ord Gi* statistic to identify statistically significant hotspots and coldspots of wildfire emissions. Additionally, the presence of spatial autocorrelation implies the need for spatially explicit regression

models (e.g., Geographically and Temporally Weighted Regression), as global models such as Ordinary Least Squares (OLS) may not adequately capture the spatial heterogeneity in emission-driver relationships.

Table 1. Global spatial autocorrelation statistics of $CO_2$ emissions

| Vegetation cover type | Moran's I | Z | p | Clustering pattern |
|---|---|---|---|---|
| Forest | 0.68 | 64.90 | 0.000 | Cluster |
| Shrub | 0.99 | 91.50 | 0.000 | Cluster |
| Grassland | 0.60 | 58.95 | 0.000 | Cluster |
| Cropland | 0.89 | 106.50 | 0.000 | Cluster |
| All | 0.60 | 83.15 | 0.000 | Cluster |

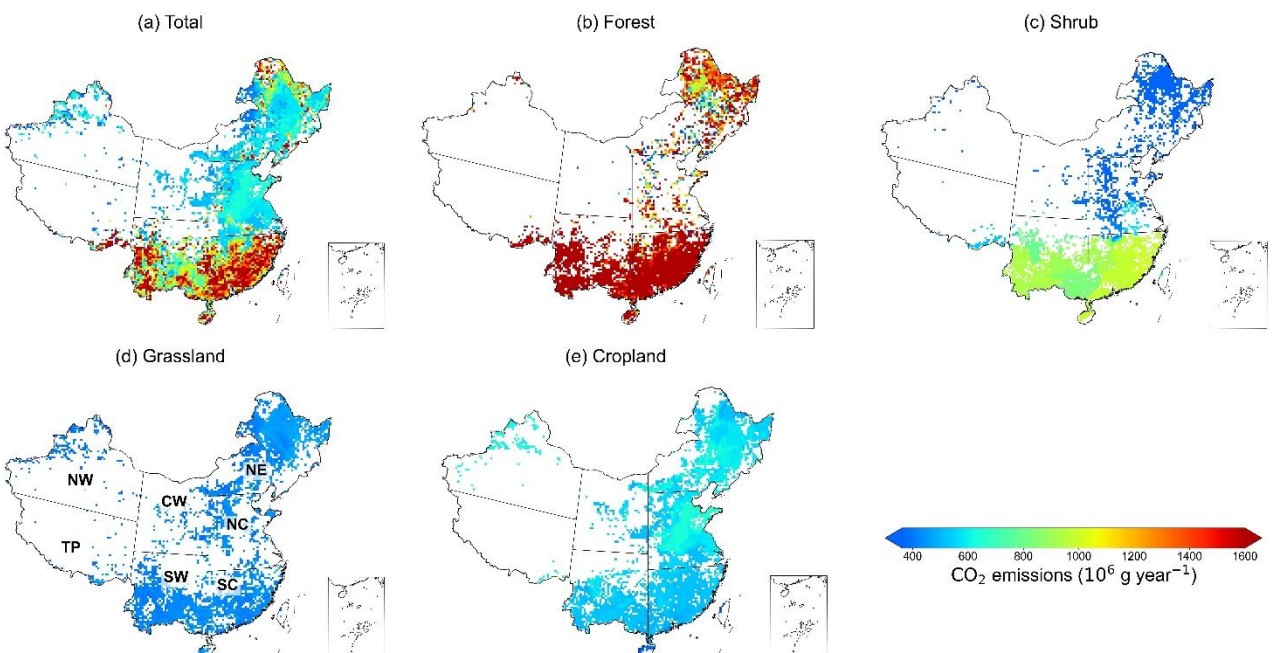

**Figure 5: Average annual spatial distribution of $CO_2$ emissions in China from 2001 to 2022: (a) all fire types, (b) Forest, (c) Shrub, (d) Grassland, and (e) Cropland. The original emission data at 500 m resolution were resampled to 0.25° grid to enhance visual clarity.**

Based on the Getis-Ord Gi* analysis (Fig. 6), we identified clear spatial clusters of persistent high (hotspots) and low (coldspots) wildfire $CO_2$ emissions (Fig. 6). Among different vegetation types, $CO_2$ emissions from forest fires are mainly distributed in NE, SW, and SC regions, with the NE region accounting for 56% of China's annual average emissions from 2001 to 2022 (Fig. 5b). The NE region (e.g., the Greater and Lesser Khingan Mountains) is a typical coniferous forest belt with abundant fuel accumulation, dry and windy spring conditions, and makes it highly prone to intense but infrequent wildfires (Lian et al., 2024a). However, despite the high forest fire emissions in NE, no significant hotspots were detected by the Getis-

Ord Gi* analysis (Fig. 6b), indicating that its high emissions mainly stem from sporadic extreme events rather than persistent clustering (Fig. 7a). For example, in 2003 and 2008, extreme wildfires in NE China contributed 73% and 56% of the national forest fire $CO_2$ emissions in 2003 and 2008, respectively (Fig. 7a). In contrast to NE, SW and SC exhibited significant spatial clustering in forest fire $CO_2$ emissions. Forest fires in these regions are prone to occurring in late winter and early spring each year, with relatively small fire scales but high frequency (Qin et al., 2014; Zhang et al., 2023a).

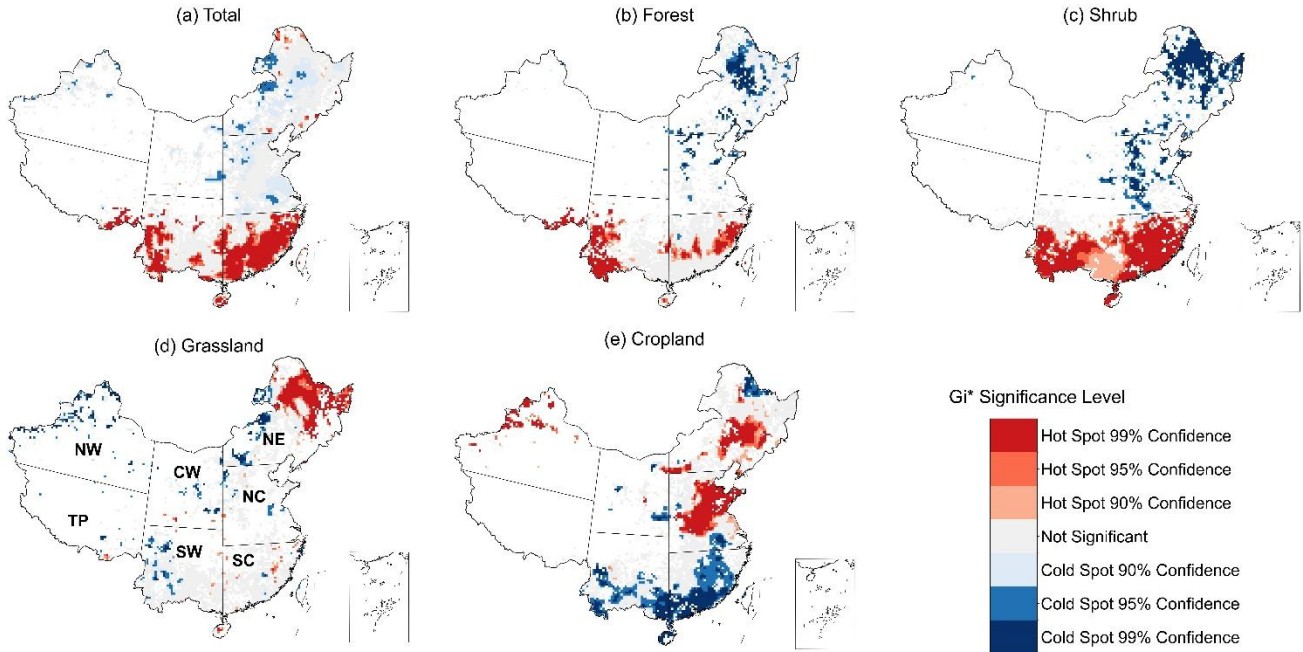

**Figure 6: Spatial Clustering of $CO_2$ emissions at 0.25° resolution in China (Gi* Hot Spot Analysis): (a) all fire types, (b) Forest, (c) Shrub, (d) Grassland, and (e) Cropland. Red areas represent statistically significant clusters of high emission values (hot spots), while blue areas indicate significant low-value clusters (cold spots), with confidence levels of 90%, 95%, and 99%.**

Shrub fire $CO_2$ emissions were concentrated in the SW and SC regions, accounting for 47% and 27% of China's annual average emissions from 2001 to 2022, respectively (Fig. 5c). Secondary vegetation such as shrubs and bamboo forests are common in these areas, resulting from land use changes (e.g., farmland abandonment, forest degradation), which facilitates the accumulation of combustibles (Han et al., 2018). Meanwhile, complex terrain and high biomass also amplify the risk of fire spread (He et al., 2024). Additionally, seasonal drought (low humidity) combined with human activities such as fuelwood collection and traditional burning practices (Ying et al., 2021) exacerbate fire occurrences, forming persistent spatial clustering that has been clearly identified as hotspot areas (Fig. 6c).

Grassland fire emissions were mainly concentrated in the NE region, accounting for 70% of the China's annual mean during 2001- 2022 (Fig. 5d), with hotspot areas focusing on the grasslands of Inner Mongolia (e.g., Hulunbuir, Xilingol) (Fig. 6d). In this region, dry herbaceous vegetation, strong winds, and low humidity in spring make grassland fires extremely prone to ignition (Chang et al., 2023). Additionally, there is a close relationship between land use and grassland fire occurrence (Li

et al., 2017). Li et al. (2017) explored the relationship between land use and the spatial distribution of grassland fires, and the results showed that land use has a significant impact on grassland fires.

High $CO_2$ emissions from cropland fires were concentrated in NC and NE regions, accounting for 51% and 42% of the China's annual mean emissions, respectively, during 2001-2022 (Fig. 5e). Spatiotemporally, from 2003 to 2012, the main emission sources were agricultural provinces in NC (e.g., Hebei, Shandong, Henan, Anhui), while after 2012, agricultural regions in NE (e.g., Heilongjiang, Jilin, Liaoning) became the primary sources of emissions (Fig. 6e and Fig. 7d). These areas have high crop straw yields and long-standing traditional burning practices, making them typical hotspots of agricultural fires (Li et al., 2024a; Wu et al., 2018).

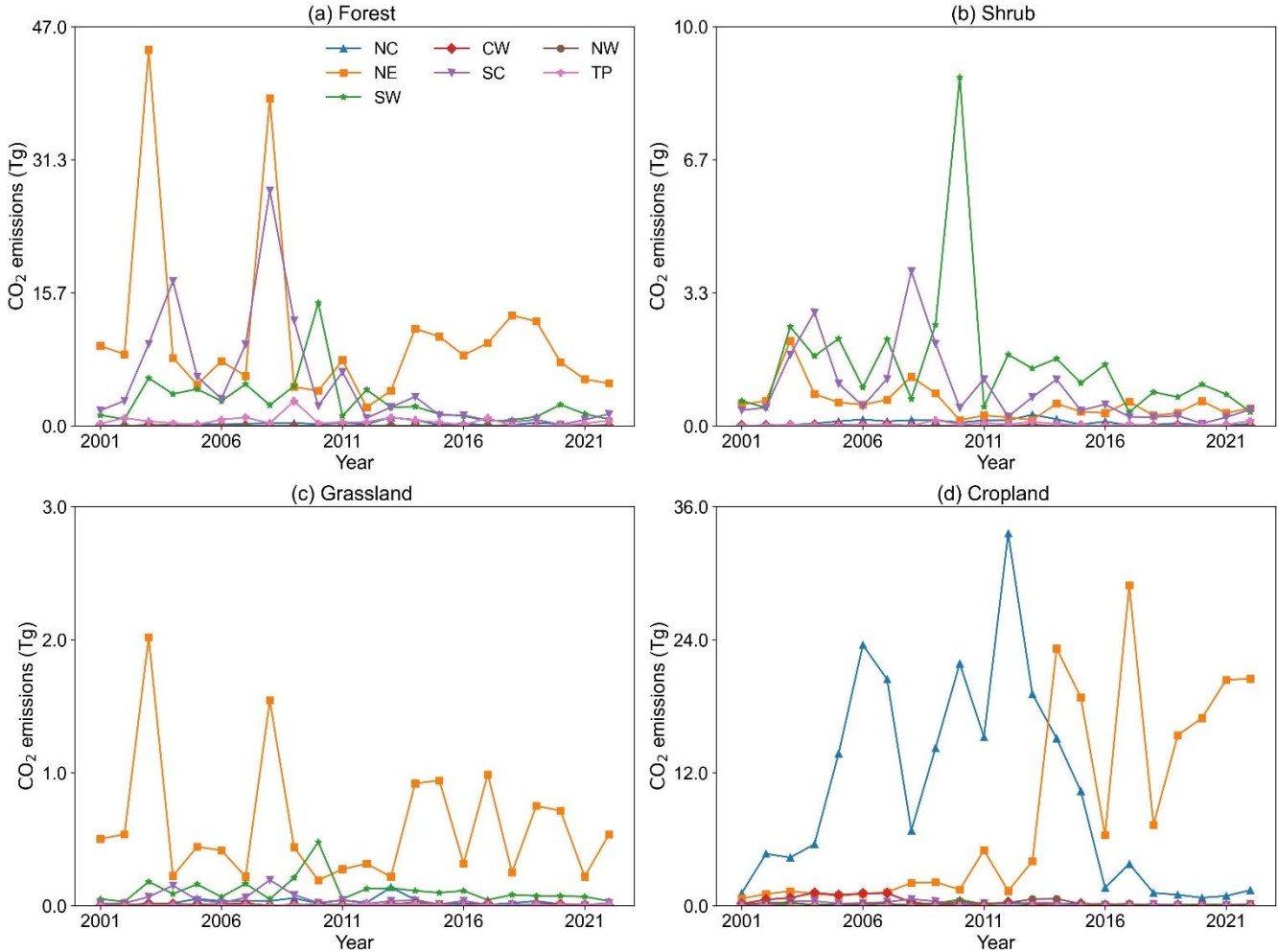

**Figure 7: Time series of $CO_2$ emissions in regions under different vegetation cover types from 2001 to 2022 in China: (a) Forest, (b) Shrub, (c) Grassland, and (d) Cropland.**

## 3.4 The impact of factors on wildfires in China

Wildfires in China exhibit distinct spatial clustering patterns. To investigate the climatic and socio-economic drivers of wildfire $CO_2$ emissions and to characterize their spatiotemporal heterogeneity, we applied three regression models—OLS, Geographically Weighted Regression (GWR), and GTWR—to four types of fires (forest, shrub, grassland, and cropland). We compared model performance using $R^2$ and the AICc. Explanatory variables were selected based on theoretical relevance and data availability for the period 2001-2019. As shown in Table 2, these variables include five climatic factors—punctual temperature (TMP, °C), accumulated precipitation (PRE, mm), relative humidity (RH, %), wind speed at 2m (WIN, m s⁻¹), and daily cumulative sunshine hours (SSD, h)—and two socio-economic indicators: gross domestic product (GDP, millionskm⁻²) and population density (POP_DEN, people grid⁻¹). These factors are widely recognized as influencing wildfire emissions (Lan et al., 2021; Ma et al., 2020; Zeng et al., 2024). TMP and SSD affect fuel flammability and combustion efficiency, PRE and RH regulate fuel moisture, and WIN promotes fire spread. Socio-economic factors reflected anthropogenic influences on fire ignition, suppression, and land-use. To ensure model parsimony and statistical robustness, variables that were not statistically significant ($p > 0.05$) in the global OLS model were excluded from subsequent GWR and GTWR analyses. All retained variables were normalized or Box-Cox transformed prior to modeling to ensure comparability.

Table 2 Driving factors and sources

| Driving factors | Abbreviation | Source |
|---|---|---|
| Punctual temperature | TMP | |
| Relative humidity | RH | Daily meteorological dataset of essential |
| Accumulated precipitation | PRE | meteorological elements of China National |
| Wind speed (2 m) | WIN | Surface Weather Station (V3.0) |
| Daily cumulative sunshine hours | SSD | |
| Gross domestic product | GDP | Chen et al. (2022) |
| Population density | POP_DEN | LandScan Global (Bright et al., 2001-2022) |

Across all fire types, both GWR and GTWR models outperformed the global OLS model. For forest fires, GTWR achieved the best performance ($R^2$ = 0.58; AICc = 128,909), while OLS explained only 6% of the variance, indicating strong spatiotemporal heterogeneity. For cropland fires, GTWR also performed well ($R^2$ = 0.52; AICc = 141,335), highlighting the influence of cropping cycles and regional factors. In shrub fires, the performance of GWR and GTWR models was nearly identical (both with $R^2$ = 0.87), and GTWR showed a slightly higher AICc (worse model fit), indicating that incorporating temporal weights did not lead to a substantial improvement. This suggests that shrub fire emissions are primarily driven by spatial heterogeneity, with limited temporal variability. For grassland fires, GTWR improved $R^2$ from 0.27 to 0.31 compared to GWR, but overall model fit remained low, indicating that other drivers—such as land use change, grazing, or local policies—play a critical role in grassland ecosystems.

Table 3. Comparison of regression results for different fire types using OLS, GWR, and GTWR models.

| Fire type | Model | Intercept | PRE | TEP | RH | WIN | SSD | GDP | POP_DEN | $R^2$ | AICc |
|---|---|---|---|---|---|---|---|---|---|---|---|
| Forest | OLS | 2698 | 49 | 727* | -259* | -864* | -100* | -2321* | -110* | 0.06 | 135428 |
| | GWR | 3260 | - | -436 | -145 | -200 | 91 | -1983 | -186 | 0.49 | 130508 |
| | GTWR | 3204 | | -524 | -205 | -161 | 126 | -469 | -888 | 0.58 | 128909 |
| Shrub | OLS | 385 | -303* | 583* | 246* | -162* | -384* | 11* | 20* | 0.27 | 104143 |
| | GWR | 733 | -31 | 111 | 100 | 5 | -61 | -10 | -1 | 0.87 | 90837 |
| | GTWR | 733 | -31 | 110 | 101 | 3 | -57 | -11 | 0 | 0.87 | 91034 |
| Grassland | OLS | 144 | 10 | -69* | 23* | -3* | -20* | -4* | 1* | 0.10 | 81654 |
| | GWR | 130 | - | -40 | 18 | -6 | -5 | -6 | -4 | 0.27 | 79908 |
| | GTWR | 131 | - | -40 | 19 | -5 | -4 | -20 | -4 | 0.31 | 79702 |
| Cropland | OLS | 239 | 25 | -12* | -44* | 14* | 80* | 2* | 0* | 0.10 | 149703 |
| | GWR | 230 | - | 11 | -31 | 16 | 2 | 50 | 3 | 0.42 | 143770 |
| | GTWR | 228 | - | 16 | -29 | 16 | 2 | 36 | 4 | 0.52 | 141335 |

Note: * An asterisk next to a number indicates a statistically significant *p*-value ($p < 0.01$).

To explore the temporal dynamics of individual variables, Figure 8 presented annual average GTWR regression coefficients for 2001-2019, revealing significant differences in how climatic and socioeconomic drivers influenced wildfire $CO_2$ emissions across vegetation cover types. Except for SSD, all other factors exhibited negative effects on forest fire $CO_2$ emissions (Fig. 8a). Temporally, regression coefficients for key variables such as POP_DEN, TMP, and GDP showed weakened negative effects after 2012, suggesting reduced sensitivity of forest fire emissions to these drivers in recent years.

This change likely reflects strengthened forest fire prevention policies and management measures implemented in China after 2012 (Fig. 12), which significantly reduced fire occurrences. Spatially, regression coefficients showed significant north-south disparities (Fig. 9). TMP and RH had a dual effect on forest fire emissions. In NE and NC regions, TMP, and RH positively correlated with forest fire emissions, indicating that warming and drying conditions may promote fire activity in temperate forests (Fang et al., 2021; Lian et al., 2024b) (Fig. 9a and 9b). In contrast, in SW and SC regions, TMP and RH exhibited

negative coefficients (Fig. 9a and 9b). This may occur because, while high temperatures can increase plant evapotranspiration and reduce fuel moisture content (Chuvieco et al., 2004), China's monsoon climate typically links high temperatures with high relative humidity, creating a threshold effect on forest fires (Ma et al., 2020). Additionally, during high-temperature periods, forest fire prevention authorities implement strict fire control measures, limiting fire occurrences (Abatzoglou et al., 2018; Hu and Zhou, 2014). GDP (Fig. 9e) showed positive effects in the NE region, but negative effects in SC and SW regions.

POP_DEN (Fig. 9f) generally displayed negative effects, especially in SW and SC regions, highlighting the role of human presence in fire suppression.

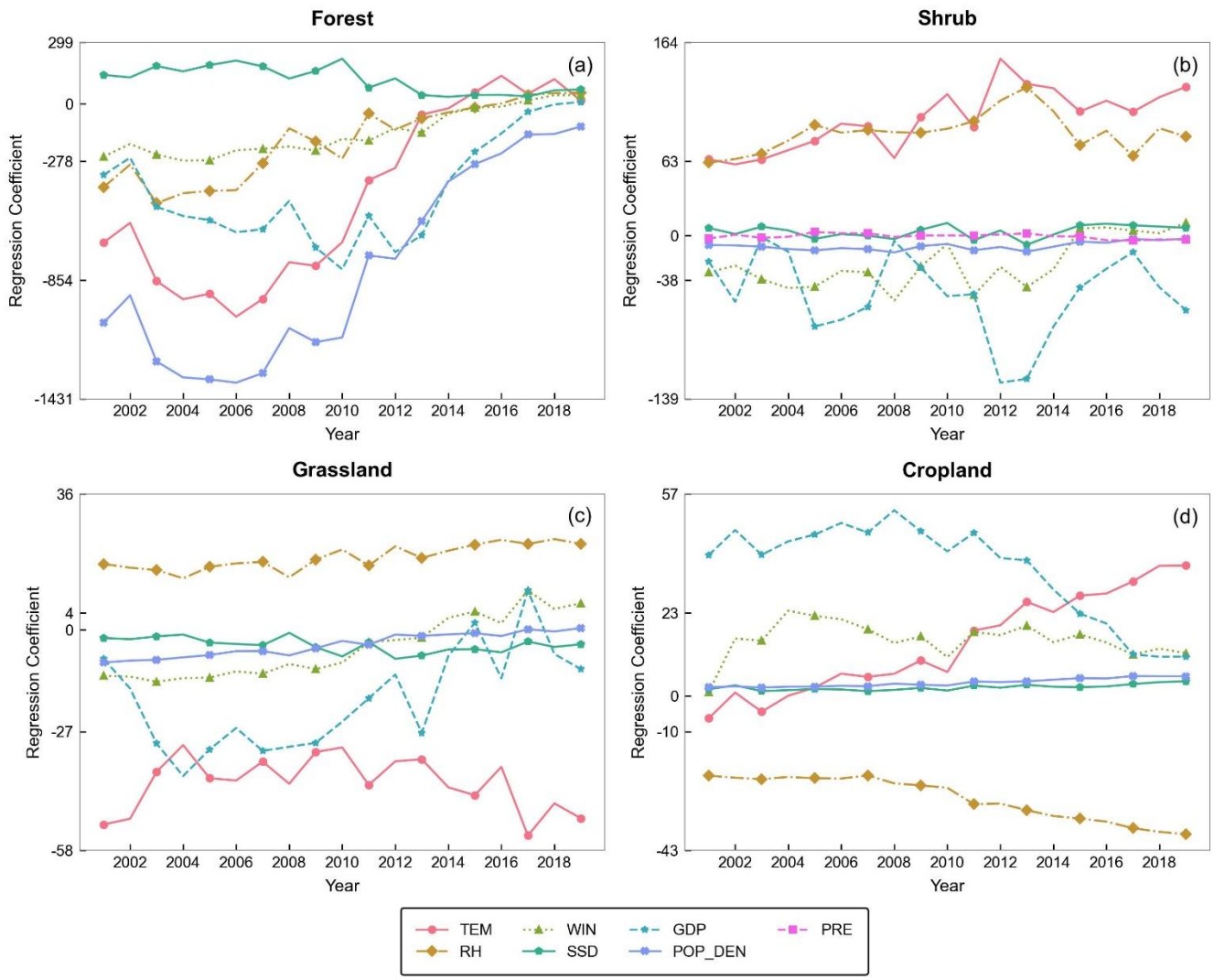

**Figure 8: Temporal evolution of GTWR regression coefficients for wildfires $CO_2$ emissions across four vegetation cover types in China (2001–2019). Positive and negative values indicate the direction and magnitude of each variable's influence on $CO_2$ emissions.**

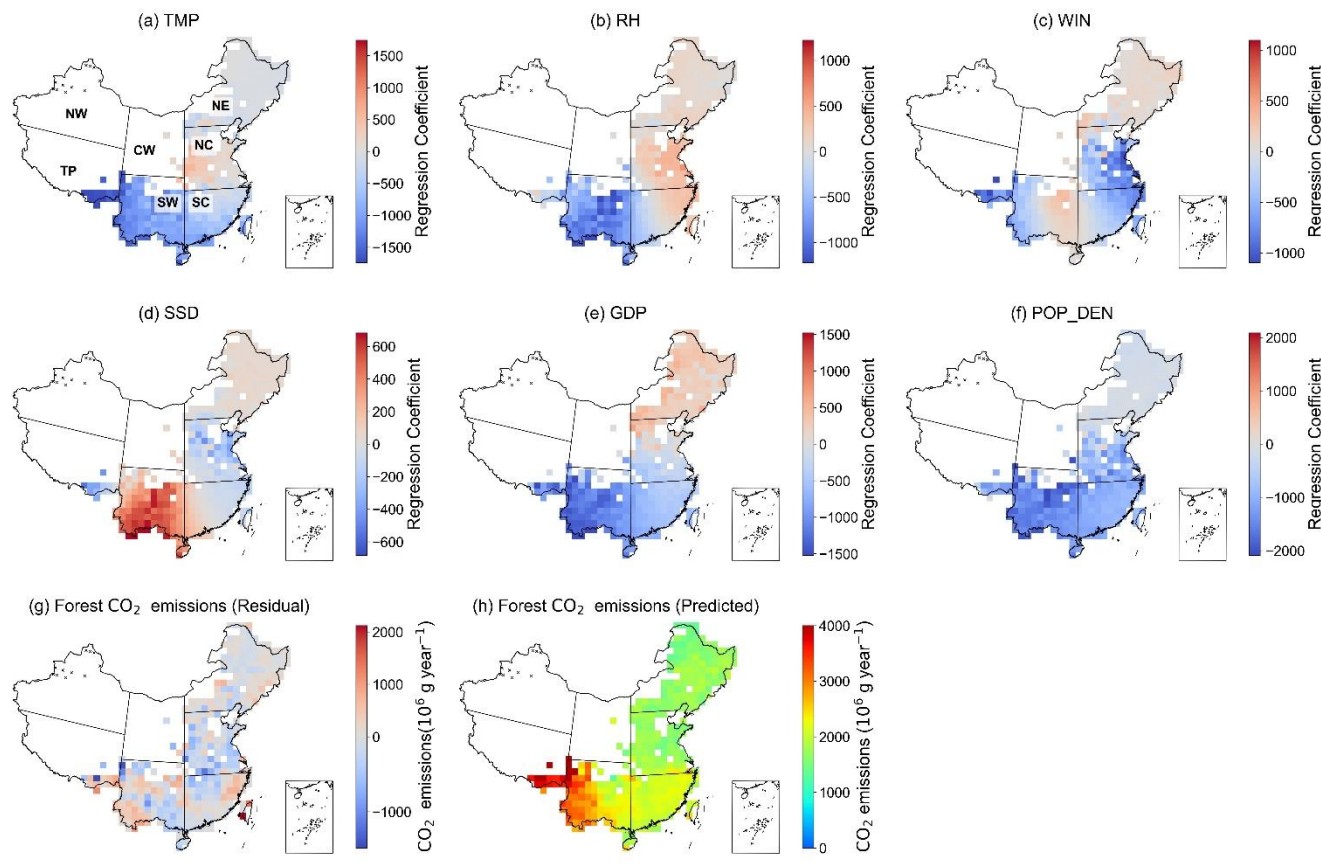

**Figure 9: Spatial distribution of GTWR regression coefficients for forest fire $CO_2$ emissions and their driving factors across China.** The maps illustrate GTWR coefficients of six environmental and socioeconomic variables: (a) temperature, (b) relative humidity, (c) wind speed, (d) daily cumulative sunshine hours, (e) gross domestic product (GDP), and (f) population density. Grey regions represent areas where the intercept was zero (i.e., no valid model fit), and black × symbols mark locations where the regression coefficients did not pass the significance test ($p \geqslant 0.05$). Figure (g) and (h) show the model residuals and predicted forest $CO_2$ emissions.

Shrub fires $CO_2$ emissions were well-captured by GWR and GTWR, dominated by spatial heterogeneity with minimal temporal variation (Table 3). GTWR showed that TMP and RH were consistent positive drivers throughout the study period, while GDP and WIN had negative effects, with other variables exerting minor influences (Fig. 8b). Spatially, TMP showed a positive effect across most regions (Fig. 10a). RH showed significant positive local effects in NC region, reflecting that humid climates promoted shrub growth and fuel accumulation (Fig. 10c) (Lian et al., 2024b; Liu et al., 2024). Once ignited by human activity or spring droughts, abundant fuel intensified fire severity and $CO_2$ emissions. In contrast, WIN and GDP exerted strong negative effects in parts of NC, likely due to effective fire control practices (Fig. 10d and 10f). Notably, a considerable number of grid cells along the northeastern border failed significance tests for at least one explanatory variable (marked as black dots in Fig. 10a-g). This may be due to limited shrub coverage or mixed land types, leading to low fire frequency and weak emission signals (Lin et al., 2024; Yang and Jiang, 2022). Additionally, many of these non-significant grids are located near international

borders, particularly adjacent to Russia. Since the GTWR model is limited to Chinese territory, therefore, it lacks information on cross-border fire activity, land use, and policy context (Li et al., 2024b; Lin et al., 2024; Quan et al., 2022). In northeastern China, transboundary fire spread is a known ignition source and may contribute to $CO_2$ emissions that are not well explained

within national data coverage.

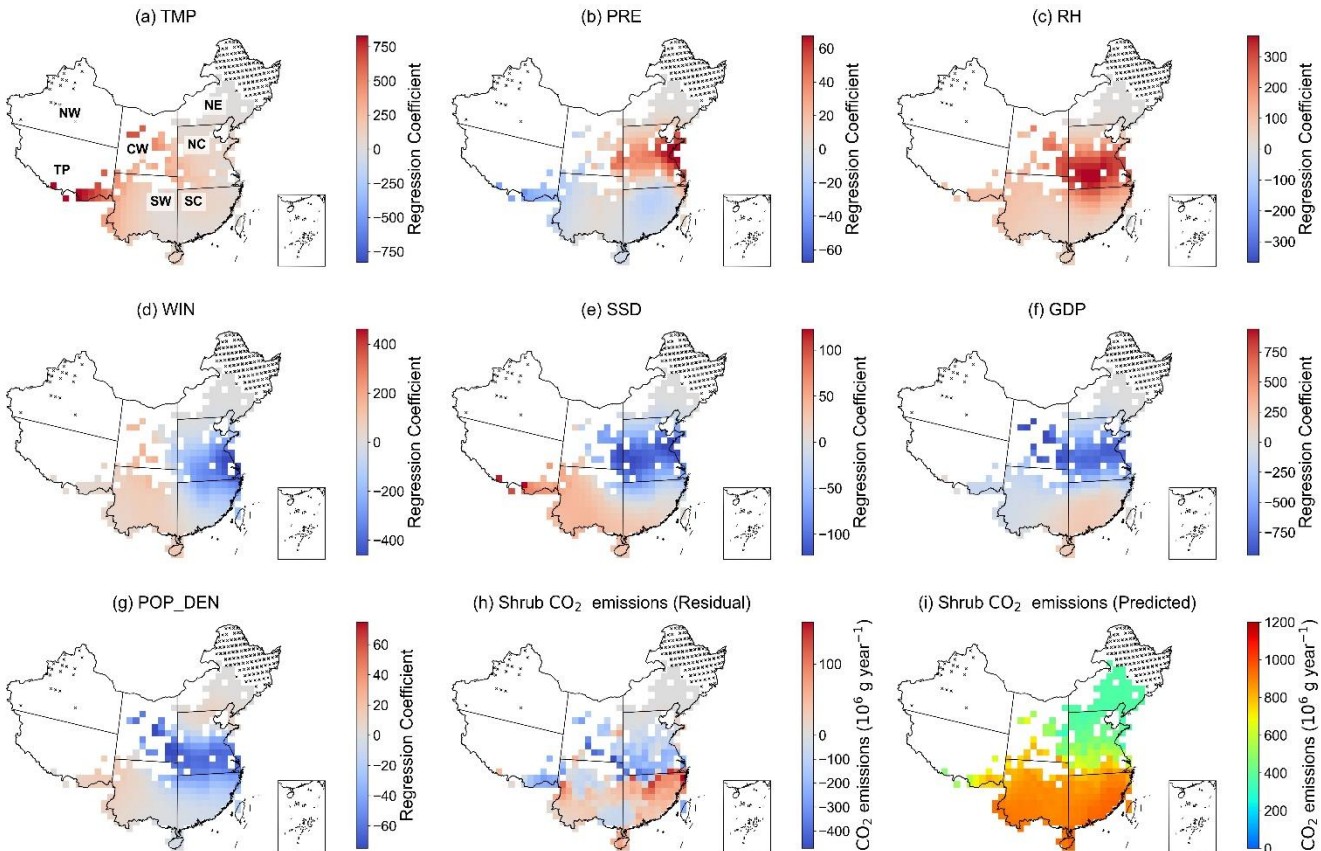

**Figure 10: Spatial distribution of GTWR regression coefficients for shrub fire $CO_2$ emissions and their driving factors across China. The maps illustrate GTWR coefficients of seven environmental and socioeconomic variables: (a) temperature, (b) precipitation, (c) relative humidity, (d) wind speed, (e) daily cumulative sunshine hours, (f) gross domestic product (GDP), and (g) population density.**

**Grey regions represent areas where the intercept was zero (i.e., no valid model fit), and black × symbols mark locations where the regression coefficients did not pass the significance test ($p \geqslant 0.05$). Figure (h) and (i) show the model residuals and predicted shrub $CO_2$ emissions.**

From a temporal perspective, the GTWR results for cropland fires $CO_2$ emissions showed that RH had a negative effect, and this negative influence gradually strengthened (Fig. 8d). GDP primarily exhibited a positive effect, but its positive

influence gradually weakened. The impact of TMP shifted from negative to positive, with its effect gradually increasing. WIN mainly exerted a positive effect, while other factors had weak influences. Spatially, TMP showed strong positive coefficients in eastern and central China (Fig. 11a). Straw burning activities in these regions peaked in spring and autumn, a pattern closely linked to rising temperatures. In contrast, negative temperature coefficients in southern and southwestern China suggest that higher temperatures in these regions, often accompanied by high humidity or stricter fire regulations, may suppress fire activity.

RH exhibited a significant negative effect across most parts of China, likely due to increased moisture content in agricultural residues, which hinders ignition and combustion (Fig. 11b). WIN showed a positive influence in CW, NC and SC regions, where expansive cropland areas may enable wind to accelerate fire spread during burning events (Fig. 11c). GDP mainly showed positive effects, but after 2010, this gradually weakened (Fig. 11e and Fig. 8d). This trend may be attributed to increased straw production driven by agricultural expansion in economically developed regions, where straw utilization

infrastructure had not yet caught up, resulting in elevated emissions. Early GDP growth likely brought more crop yields and straw generation, thereby enhancing $CO_2$ emissions(Ren et al., 2019). However, after 2012, this trend reversed as nationwide straw burning bans were introduced. Regions with higher economic development began to demonstrate stronger emission control capacity, leading to a gradual weakening of GDP's positive effect on emissions (Zeng et al., 2024).

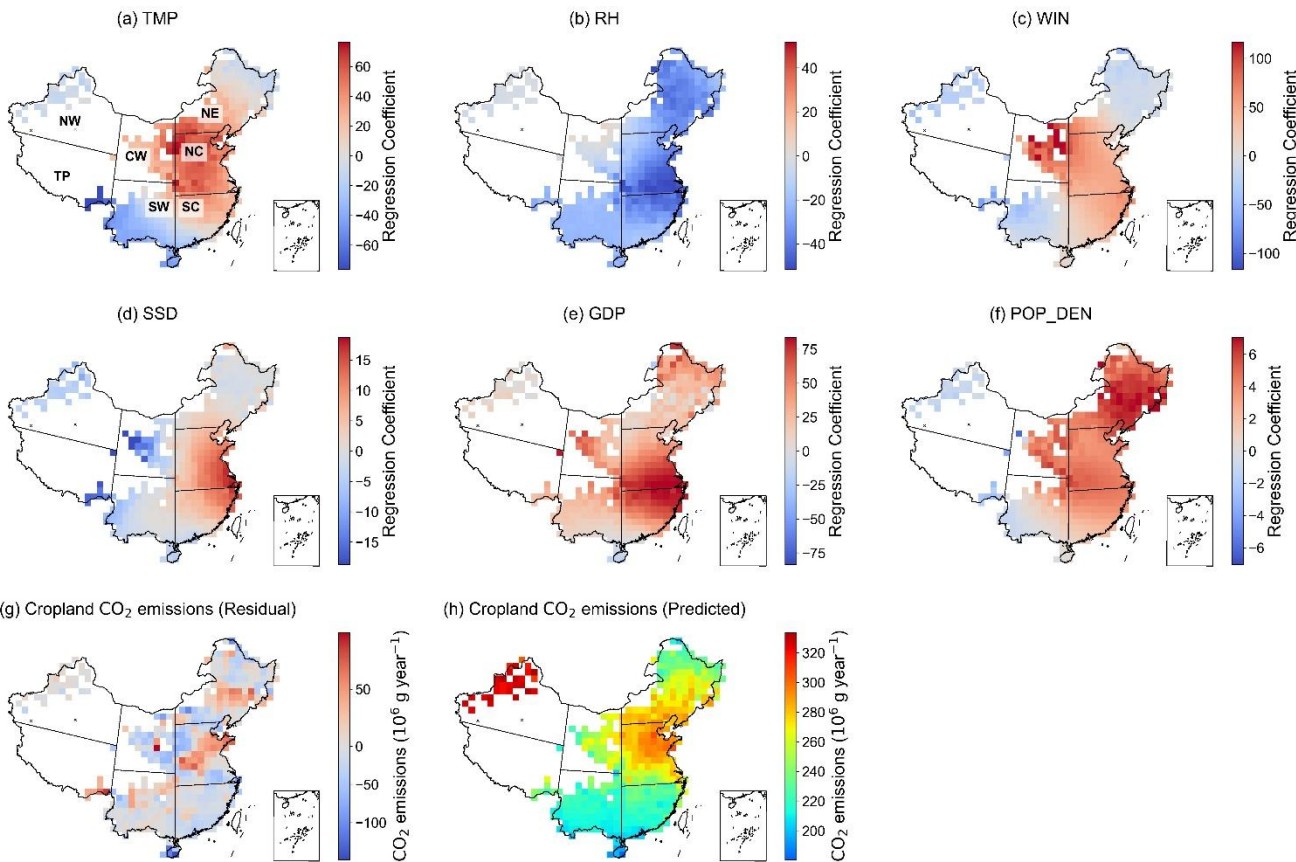

**Figure 11: Spatial distribution of GTWR regression coefficients for cropland fire $CO_2$ emissions and their driving factors across China. The maps illustrate GTWR coefficients of six environmental and socioeconomic variables: (a) temperature, (b) relative humidity, (c) wind speed, (d) daily cumulative sunshine hours, (e) gross domestic product (GDP), and (f) population density. Grey regions represent areas where the intercept was zero (i.e., no valid model fit), and black × symbols mark locations where the regression coefficients did not pass the significance test ($p \geqslant 0.05$). Figure (g) and (h) show the model residuals and predicted**
**cropland $CO_2$ emissions.**

Although the GTWR model indicated that climatic and socioeconomic variables such as TMP, RH, and GDP explained the spatial variation in wildfire $CO_2$ emissions, the overall model performance remains moderate for forest and shrub fires, with particularly low explanatory ability for grassland fires ($R^2 = 0.31$). This gap suggests that, beyond natural and socioeconomic factors, other key drivers may have been omitted. Multiple studies (Gao et al., 2023; Kelly et al., 2013; Phillips et al., 2022; Xie et al., 2020) highlight the substantial impact of fire management policies on $CO_2$ emissions. For instance, Phillips et al. (2022) showed that the marginal abatement cost of avoiding fire-related $CO_2$ emissions through fire management is comparable to or even lower than that of many other climate mitigation strategies.

In China, the role of policy is particularly significant. Wu et al. (2018) reviewed 51 crop straw management regulations issued between 1965 and 2015, with 34 implemented after 2008. The timing of these intensive regulatory efforts closely aligns with key turning points in emission trends (Fig. 12). For cropland fires, annual $CO_2$ emissions increased from 8.2 Tg year$^{-1}$ during 2001-2005 to 26.2 Tg year$^{-1}$ during 2010-2016, but began to decline following the revision of the Air Pollution Prevention and Control Law in 2015 and the launch of the Air Pollution Action Plan in 2013. Similarly, after the implementation of the National Forest Fire Prevention Plans in 2009 and 2016, $CO_2$ emissions from forest, shrub and grassland fires dropped from 38.1 Tg year$^{-1}$ (2006-2009) to 13.3 Tg year$^{-1}$ (2017-2022). Jin et al. (2022) further estimated that over 80% of wildfire-related $CO_2$ emissions could be avoided under effective fire management. These findings strongly indicate that policy management plays a critical role in wildfire $CO_2$ emissions.

Notably, northeastern China is the only region where cropland burning has increased in recent years, highlighting the need for adaptive rather than restrictive policies. As one of China's major grain-producing regions, Northeast China generates large volumes of straw. Harsh winters and short windows for straw return or removal, combined with long-established farming practices, have made complete bans on straw burning particularly challenging. Prior to strict open-burning prohibitions, farmers often burned straw in a dispersed, low-intensity manner, making detection by satellite-based fire products difficult, potentially resulting in systematic underestimation of early emissions. After the implementation of strict bans, facing growing pressure from unprocessed straw accumulation, therefore, some local governments adopted more adaptive fire management policies, such as designating burning windows under favorable meteorological conditions. These "limited and concentrated burning periods" led to spatiotemporally clustered fire events that were more easily captured by remote sensing. In recent years, the Chinese government has also promoted the scientific incorporation of straw into soils, off-field collection, and the industrial utilization of crop residues in Northeast China. These efforts highlight the significant role of policy in shaping emission trends from agricultural burning, particularly in regions where environmental constraints and traditional farming practices pose unique challenges.

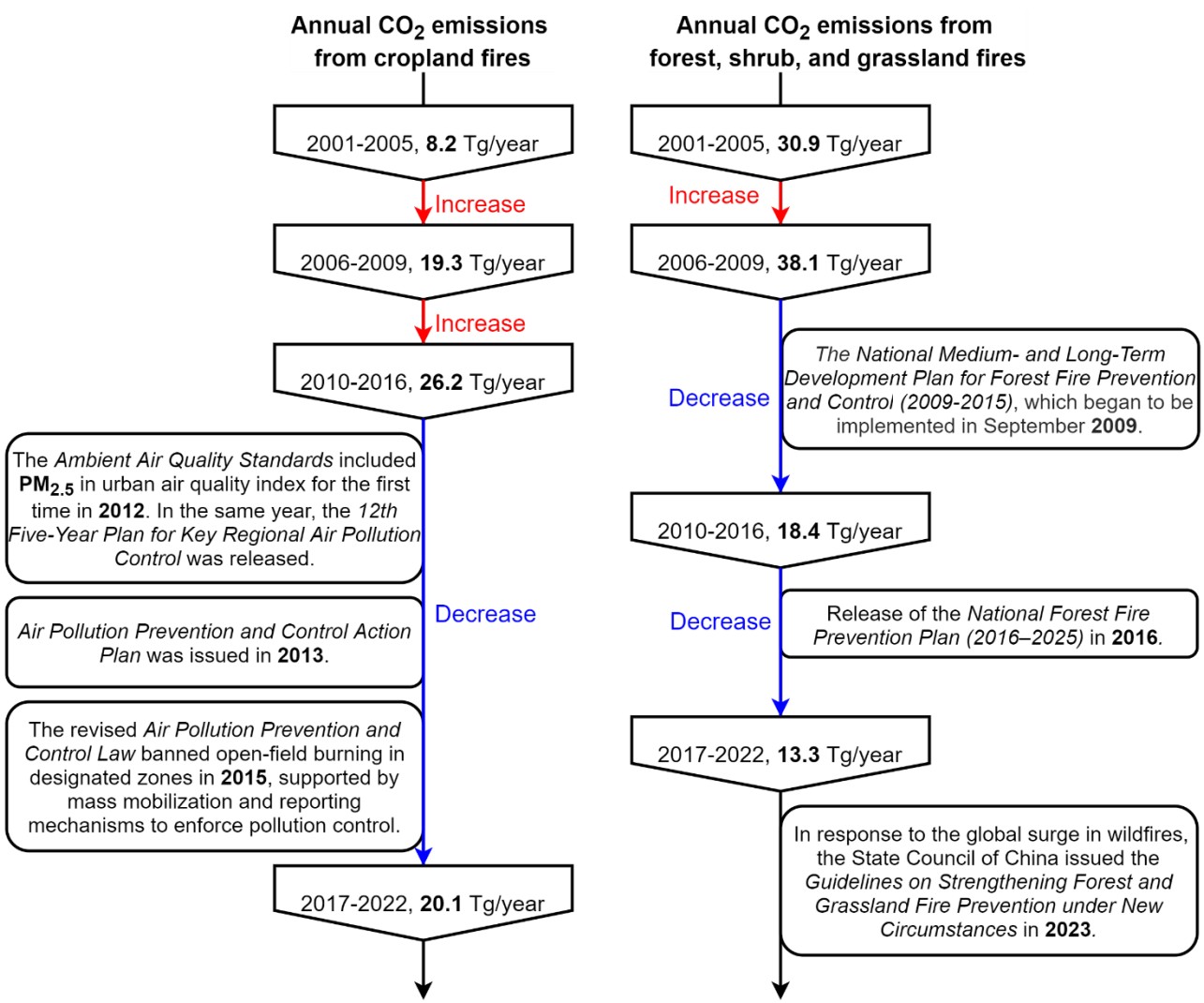

**Figure 12. Temporal trends in annual CO$_2$ emissions from cropland burning and forest, shrub, and grassland fires in China (2001–2022), and key national policy milestones related to fire and air pollution control.**

### 3.5 Uncertainty analysis

A Monte Carlo simulation (100,000 iterations) was conducted to assess the uncertainty in the estimated wildfire CO$_2$ emissions. Monte Carlo simulation is a probabilistic method that generates a large number of possible outcomes based on random sampling from the input parameter distributions, thereby providing a comprehensive assessment of model uncertainty. The uncertainties in emission estimates in this study mainly originated from satellite-derived BA products, AGB, CE, and EF. All parameters, except CE, were assumed to follow normal distributions, as suggested by Zhao et al. (2011). CE values were

assigned triangular distributions based on vegetation types, with parameter ranges derived from empirical data and literature sources (Junpen et al., 2020; Mieville et al., 2010; Ping et al., 2021; Van Leeuwen et al., 2014; Zhou et al., 2017). For forest and grassland fires, CE was parameterized the FVC-based empirical relationship proposed by Hély et al. (2003), while fixed CE values were applied to shrub and cropland fires. The coefficients of variation (CV) of EF were estimated based on the mean and variability summarized from multiple published sources. Monte Carlo simulations showed that CE and EF contributed less to total emission variability compared to BA and AGB.

Among all parameters, BA emerged as the dominant source of uncertainty. However ,the uncertainty in retrieved from satellite products is difficult to quantify (Hoelzemann et al., 2004; Wu et al., 2018). The MCD64A1 product performs reliably in detecting large fires (Giglio et al., 2018), and its CV was adopted from Giglio et al. (2010). While we also recognize that the MODIS MCD64A1 product tends to underestimate small, fragmented, or low-intensity fires. To evaluate and adjust for this underestimation, we conducted a comprehensive comparison using FireCCI51 (250m resolution), GFED (500m resolution), the novel 30-m resolution Global annual Burned Area Map (GABAM, 30m resolution) (Long et al., 2019), and FINN datasets for the year 2015 (Table S5). The comparison showed that MODIS systematically underestimated burned areas. Despite its higher spatial resolution, GABAM reported smaller cropland fire areas, likely due to its limited temporal resolution. The FINN dataset differed significantly from all other products, with its burned areas generally higher than other data products. Based on these comparisons, we derived a scaling factor ($\alpha_i$) using the FireCCI51 and GFED datasets and applied them to MODIS burned area estimates. On average, this adjustment increased MODIS-based BA estimates by approximately 1.5 times. To further evaluate the representativeness of our correction method, we compared the standard FINN dataset with a revised version, FINN_VIIR, which incorporates VIIRS active fire detection data (375m resolution). VIIRS is known to better capture small and short-duration fires often missed by MODIS. Our analysis showed that the burned area in FINN_VIIR was approximately 40% higher than in the standard FINN dataset, which closely aligns with the scaling factor applied in our MODIS-based correction. This consistency provides further support for the effectiveness of our BA adjustment strategy.

AGB is another major contributor to emissions uncertainty. To reflect interannual changes in biomass, we employed the AGB dataset from Su et al. (2016) for the period 2001-2012 and that from Yin et al. (2023) for 2013-2022 for calculating forest fire $CO_2$ emissions. The mean difference between the two datasets was approximately 7% (100 t ha$^{-1}$ vs. 107 t ha$^{-1}$), well within the ±50% uncertainty range reported by Yin et al. (2023), confirming their compatibility for long-term analysis. For shrub fire $CO_2$ emissions, we employed localized biomass density values from Hu et al. (2006), enhancing the regional representativeness of AGB inputs. For grassland fire $CO_2$ emissions, we used the index model based on the NDVI developed by Gao et al. (2006). We acknowledge that NDVI-based models may underestimate AGB in dense vegetation due to saturation. To address this, we compared the exponential model by Gao et al. (2012) with the saturation-corrected model by Hu et al. (2024) for alpine meadows in China. The mean AGB estimates were 210 g m$^{-2}$ (Gao et al., 2012) and 214 g m$^{-2}$ (Hu et al., 2024), with a small difference of 1.9%, well within the reported uncertainty bounds of both models (±62.5 g m$^{-2}$ for Gao et al., 2012; ±85 gm$^{-2}$ for Hu et al., 2024). Given the broader applicability of Gao's model across diverse grassland types (e.g., arid steppe, wetlands, meadow grasslands), we adopted it for national-scale grassland AGB estimation. For forest, shrub and

grassland fire $CO_2$ emissions, the uncertainty in AGB was derived from values reported in the literature. For cropland fire $CO_2$ emissions, AGB was derived from national statistical records, with CV set at 20% (Zhou et al., 2017).

Table 4 presented the total wildfire $CO_2$ emissions and their associated uncertainty ranges across different vegetation cover types. On average, the estimated uncertainties in $CO_2$ emissions were (-39%, +76%) for forest fires, (-37%, +20%) for shrub fires, (-26%, +58%) for grassland fires, and (-50%, +51%) for cropland fires. The large uncertainties in forest, shrub, and grassland fire $CO_2$ emissions were mainly due to uncertainties in AGB and BA estimates. The uncertainty in cropland fire $CO_2$ emissions uncertainty primarily reflected possible under-detection of BA. Despite these uncertainties, this study

incorporated multiple BA datasets, multi-temporal vegetation cover datasets, regionally validated AGB estimates, and a comprehensive set of EF, resulting in a spatially representative characterization of wildfire $CO_2$ emissions and their temporal evolution in China.

Table 4 The uncertainty estimation of wildfires $CO_2$ emissions from 2001 to 2022.

| Year | Forest | Shrub | Grassland | Cropland | All types |
|------|--------|-------|-----------|----------|-----------|
| 2001 | (-38%, 72%) | (-35%, 18%) | (-28%, 62%) | (-49%, 39%) | (-39%, 62%) |
| 2002 | (-35%, 68%) | (-34%, 16%) | (-24%, 47%) | (-49%, 34%) | (-39%, 54%) |
| 2003 | (-39%, 74%) | (-34%, 15%) | (-35%, 65%) | (-51%, 47%) | (-39%, 66%) |
| 2004 | (-36%, 69%) | (-37%, 22%) | (-15%, 58%) | (-50%, 40%) | (-39%, 57%) |
| 2005 | (-31%, 58%) | (-33%, 14%) | (-18%, 57%) | (-54%, 45%) | (-41%, 47%) |
| 2006 | (-30%, 56%) | (-32%, 12%) | (-18%, 62%) | (-56%, 47%) | (-46%, 48%) |
| 2007 | (-29%, 54%) | (-32%, 13%) | (-13%, 48%) | (-53%, 34%) | (-40%, 41%) |
| 2008 | (-50%, 106%) | (-42%, 29%) | (-39%, 67%) | (-52%, 43%) | (-50%, 93%) |
| 2009 | (-38%, 74%) | (-39%, 26%) | (-20%, 56%) | (-51%, 35%) | (-42%, 54%) |
| 2010 | (-37%, 73%) | (-45%, 34%) | (-15%, 82%) | (-59%, 50%) | (-48%, 56%) |
| 2011 | (-36%, 68%) | (-36%, 19%) | (-23%, 53%) | (-51%, 42%) | (-44%, 52%) |
| 2012 | (-35%, 66%) | (-36%, 18%) | (-27%, 57%) | (-58%, 44%) | (-52%, 47%) |
| 2013 | (-35%, 69%) | (-33%, 14%) | (-19%, 44%) | (-48%, 34%) | (-43%, 42%) |
| 2014 | (-40%, 79%) | (-36%, 19%) | (-26%, 68%) | (-43%, 39%) | (-41%, 50%) |
| 2015 | (-40%, 76%) | (-39%, 25%) | (-35%, 53%) | (-44%, 43%) | (-42%, 53%) |
| 2016 | (-44%, 91%) | (-42%, 29%) | (-24%, 54%) | (-46%, 56%) | (-44%, 70%) |
| 2017 | (-40%, 76%) | (-37%, 21%) | (-37%, 57%) | (-45%, 59%) | (-43%, 62%) |
| 2018 | (-53%, 114%) | (-41%, 27%) | (-30%, 59%) | (-45%, 62%) | (-49%, 90%) |
| 2019 | (-43%, 83%) | (-34%, 16%) | (-36%, 50%) | (-49%, 72%) | (-45%, 74%) |
| 2020 | (-48%, 95%) | (-40%, 24%) | (-39%, 62%) | (-53%, 86%) | (-50%, 85%) |

| 2021 | (-45%, 89%) | (-38%, 21%) | (-28%, 53%) | (-57%, 98%) | (-53%, 92%) |
| 2022 | (-35%, 66%) | (-34%, 16%) | (-24%, 56%) | (-45%, 66%) | (-42%, 63%) |

**3.6 Comparison with other studies**

We compared the wildfire $CO_2$ emissions estimates in this study with several global biomass burning inventories, including the Fire Inventory from NCAR (FINN2.5, FINN_VIIRS2.5) (Wiedinmyer et al., 2023); GFED5 (van der Werf et al., 2017), the Global Fire Assimilation System (GFAS version 1.2) (Kaiser et al., 2012), and the Quick Fire Emissions Dataset (QFED version 2.5) (Koster and Darmenov, 2015), as shown in Figure 13. While all inventories exhibited consistent interannual variability, the total emission magnitudes varied substantially. Our estimates were systematically lower than those

from most global datasets and were closest to GFASv1.2. FINN2.5 reported the highest values among all inventories, likely due to its use of larger burned area inputs. By incorporating VIIRS active fire detections, FINN_VIIRS2.5 showed approximately 25% higher emissions than FINN2.5. We applied GFED5 burned area data to adjust MCD64A1 burned area estimates; however, our emissions remain lower than those from GFED, which may be attributed to differences in biomass assumptions. Biomass input remains a dominant source of uncertainty in fire emissions estimates. QFED2.5 adopted a top-

down approach based on fire radiative energy (FRE), and typically yields higher emission estimates (Wiedinmyer et al., 2023; Yin et al., 2019).

We further compared our estimates with other studies in China (Table 5). For forest, shrub, and grassland fires, our $CO_2$ estimates were comparable to those reported by Zhou et al. (2017) and Li et al. (2024a), slightly higher than those of Jin et al. (2022), and lower than Yin et al. (2019). The lower values in Jin et al. (2022) may result from exclusive reliance on MODIS

burned area data, whereas the higher values in Yin et al. (2019) stem from the use of the FRE-based method. Regarding cropland fires, remote sensing often fails to detect small-scale agricultural burning. Consequently, many studies have used statistical data to estimate emissions, based on assumed field residue burning percentages ranging from 10% to 80% (Gao et al., 2002; Huang et al., 2012; Li et al., 2024b; Wang and Zhao, 2008; Yan et al., 2006; Yang et al., 2008; Zhou et al., 2017). The cropland fire $CO_2$ emissions estimated by the method based on burning proportion are generally higher than those

calculated by the satellite remote sensing monitoring method adopted in this study.

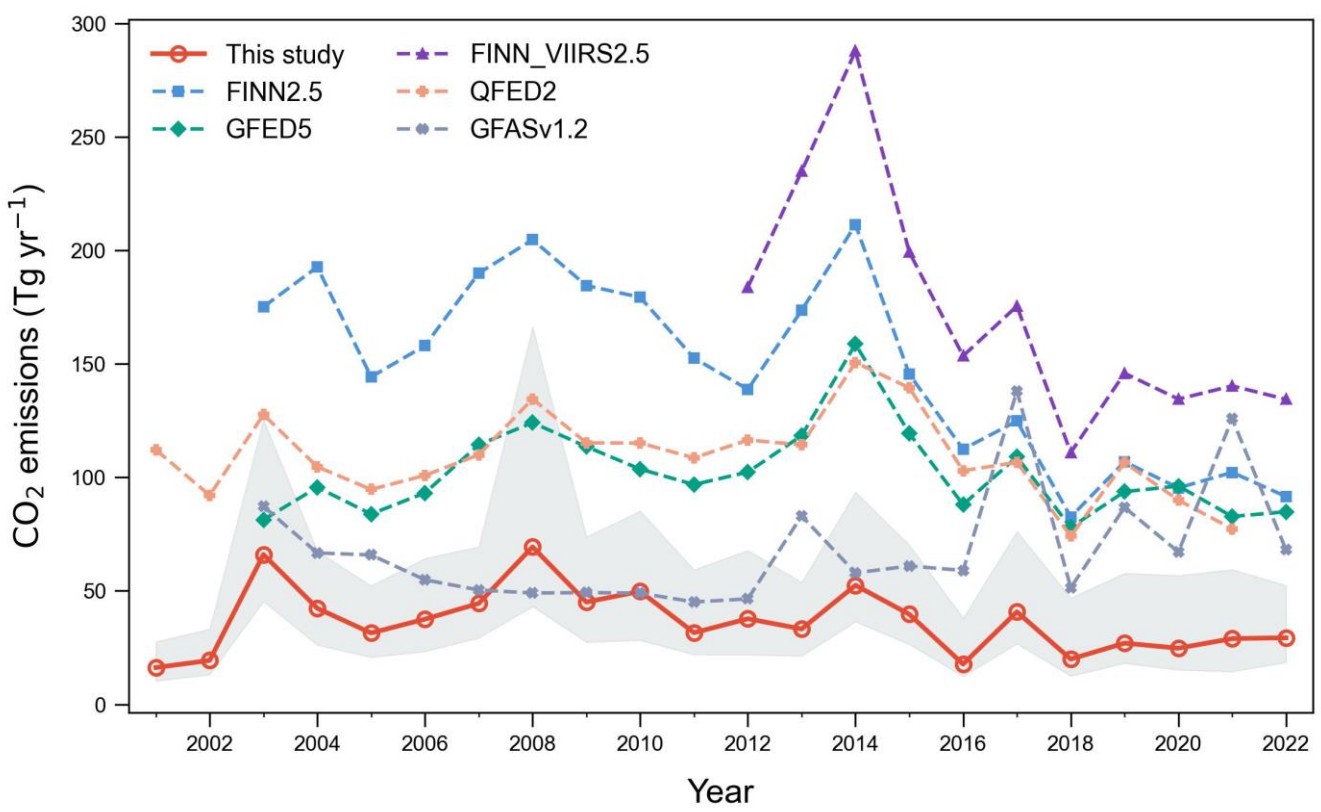

**Figure 13: Comparison of wildfires CO$_2$ emissions in China from multiple global inventories (2001–2022).**

Table 5 Comparison of wildfires CO$_2$ emissions estimates in China from previous studies (Tg).

| Reference | year | region | Forest | Shrubland | Grassland | Cropland | All types |
|---|---|---|---|---|---|---|---|
| Wang and Zhao. (2008) | 2006 | China | - | - | - | 154.5 | - |
| Huang et al. (2012) | 2006 | China | - | - | - | 68 | - |
| This study | 2006 | China | - | - | - | 26.3 | - |
| Zhou et al. (2017) | 2012 | China | | 10.1 | 0.7 | 207.3 | - |
| This study | 2012 | China | | 8.0 | 2.4 | 35.9 | - |
| Yin et al.(2019) | 2003-2017 (mean) | China | 40.8 | 1.2 | 14.1 | 35.3 | 91.4 |
| This study | 2003-2017 (mean) | China | 23.1 | 4.0 | 0.9 | 21.7 | 49.7 |

| | | | | | | | |
|---|---|---|---|---|---|---|---|
| Jin et al.(2022) | 2001-2019 (mean) | China | 15.2 | | | - | - |
| This study | 2001-2019 (mean) | China | 22.2 | | | - | - |
| Li et al.(2024a) | 2001-2020 (mean) | Heilongjiang | 3.9 | - | 0.1 | 13.2 | - |
| This study | 2001-2020 (mean) | Heilongjiang | 4.7 | 0.3 | 0.3 | 5.0 | - |

## 4 Conclusion

This study developed a comprehensive inventory of wildfire $CO_2$ emissions across China from 2001 to 2022, capturing significant spatiotemporal variations among different vegetation types. Results showed that cropland and forest fires were the primary contributors to national wildfire emissions. Forest and shrub fire $CO_2$ emissions exhibited a declining trend, grassland fire $CO_2$ emissions remained relatively stable, and cropland fire $CO_2$ emissions showed an increasing trend. GTWR analysis revealed that shrub fire $CO_2$ emissions exhibited the highest predictive performance ($R^2 = 0.87$), with climatic factors

(particularly temperature and humidity) being the main influencing factors, and limited temporal variation. In contrast, forest and cropland fire $CO_2$ emissions were significantly influenced by the spatiotemporal heterogeneity of both climatic and socioeconomic factors. Grassland fire $CO_2$ emissions exhibited the lowest model explanatory power ($R^2 = 0.31$), suggesting that their emissions may largely depend on drivers not included in the current model.

    Our findings underscore the critical role of policy interventions in shaping wildfire emissions in China. The observed

declines in most regions aligned with the implementation of national fire control and air pollution reduction programs. However, northeastern China remained an exception, with cropland fire $CO_2$ emissions continuing to increase in recent years. This trend highlighted the limitations of blanket burning bans and the necessity of adaptive fire management. Although forest fire $CO_2$ emissions had been reduced through strengthened fire prevention measures, northeastern China remained vulnerable to extreme fire events triggered by drought or lightning. Shrub fire $CO_2$ emissions, primarily driven by climatic factors,

underscore the importance of strengthening early-warning systems.

    Although wildfire emissions are classified as "natural disturbances" under IPCC guidelines for LULUCF and are often excluded from national emission inventories, the results demonstrated that these emissions were substantial and closely tied to policy and land management practices. The pronounced interannual variability and spatial heterogeneity suggested that future climate extremes, land-use changes, or fire policy adjustments could significantly alter regional carbon dynamics.

Compared with global emission inventories (GFED, FINN, QFED, GFAS), the estimates in this study were generally lower. Although remote sensing data might underestimate some cropland fires, this study characterized wildfire $CO_2$ emissions patterns in China by integrating multi-source burned area products, localized biomass data, and high-resolution land cover

classifications. Future research should further refine burned area identification, optimize parameters such as emission factors and combustion efficiency, bridge observational gaps, and incorporate transboundary fire dynamics to ensure more comprehensive and accurate regional emission accounting.

## Acknowledgments

This work was supported by the National Natural Science Foundation of China (42221003 and 41991250), the Strategic Priority Research Program of Chinese Academy of Sciences (XDB40000000).

## Data availability

All the data supporting the findings of this paper can be accessed via the provided links or by requesting them using the contact information provided within those links. The China Land Use Land Cover Remote Sensing Monitoring Dataset (CNLUCC) is sourced from the Resource and Environment Science Data Registration and Publishing System (https://www.resdc.cn/, last access: 25 May 2025). China's regional 250 m fractional vegetation cover dataset, China's regional 250 m normalized difference vegetation index dataset, and The Daily Meteorological Dataset of Essential Meteorological Elements of the China National Surface Weather Station (V3.0) are sourced from the National Tibetan Plateau/Third Pole Environment Data Center (https://data.tpdc.ac.cn/, last access: 25 May 2025). MODIS-MCD64A1 burned area data are publicly available at https://lpdaac.usgs.gov/products/mcd64a1v061 (last access: 25 May 2025). A 1 km harvesting area dataset is available at http://dx.doi.org/10.17632/jbs44b2hrk.2 (last access: 25 May 2025). The FireCCI51 burned area product is publicly available from the European Space Agency Climate Change Initiative (ESA CCI) at http://cci.esa.int/data (last access: 25 May 2025). The GFED500 product can be accessed via the GFED website at https://www.globalfiredata.org/ (last access: 25 May 2025). The LandScan 2020 Global Population Database is available at https://landscan.ornl.gov/ (last access: 25 May 2025).

## Author contributions

The article was written with contributions from all the authors. Xuehong Gong and Yongming Han designed this study; Xuehong Gong, Zeyu Liu, Jie Tian, and Qiyuan Wang collected and organized the data; Xuehong Gong analyzed the data and wrote the original article with contributions from Zeyu Liu, Yongming Han, Guohui Li, and Zhisheng An. Zeyu Liu, Jie Tian, and Qiyuan Wang assisted with the submission of the article. All the authors have given approval to the final version of the article.

## Competing interests

The contact author has declared that none of the authors has any competing interests.

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
