# Peer review of "Spatiotemporal patterns and drivers of wildfire CO2 emissions in China from 2001 to 2022"

_EGUsphere, 2024_

## Author Response (AR1)

**Responses to the reviewer comments**

**Reviewer #1:**

**General comments**: Wildfire emissions are challenging to quantify but often impact air quality and climate. The study by Gong et al. 2024 estimates China's $CO_2$ emissions caused by wildfires from 2001-2022, finding high emission potential from cropland and forests. The authors use the bottom-up method using MODIS-MCD64A1 burned area product, Chinese LULC (CNLUCC) product, and NDVI product in conjunction with ancillary datasets such as harvesting area map, meteorological variables and a set of literature-based emission factors to assess changes in fire emissions across China for nearly two decades. The study highlights a declining trend of forest emissions while cropland fires have risen over the years. Further, statistical correlations of fire emissions with sunshine hours, NDVI, and temperature are analysed to determine the dominant influencing factors of the forest and shrub fires.

From a research perspective, however, the current version of the manuscript lacks certain major aspects (outlined in detail below), which require careful evaluation of results, such as the comparison with existing inventory and estimations of the emission uncertainty that may arise from both data and methods. The result interpretation and discussion are found weak on a number of occasions, limiting the scientific scope of the article - the study may need to provide more evidence (direct or indirect) to back up the interpretations of the results. The overall structure of the paper could be improved by adding the novelty of the present study and avoiding disjoints (detailed below). Only after adequately addressing these major issues can I recommend this study for a potential publication in ACP.

**Response**: We sincerely thank the reviewer for the comprehensive and constructive evaluation of the manuscript. In response to your main concerns, the following revisions have been made. In the Results and Discussion section (Section 3.6, "Comparison with other studies"), we compared our findings with global fire emission inventories (e.g., GFED, FINN, QFED) and regional studies in China. This contextual comparison provides a reference for our estimates and enhances the interpretability of the emission inventory. The revised manuscript further includes a Monte Carlo uncertainty analysis, detailed in Section 3.5 "Uncertainty analysis". The results of this uncertainty analysis are summarized in Table 4, which lists the mean wildfire $CO_2$ emissions and corresponding 5th-95th percentile ranges for each vegetation cover type. The discussion section was revised to include spatial regression models (GWR, GTWR) for evaluating the drivers of emission patterns, controlling for multicollinearity and spatial dependence (Section 3.4 "The impact of factors on wildfires in China"). Notably, the GTWR model revealed strong spatiotemporal heterogeneity in the relationships between climate/socioeconomic factors and wildfire $CO_2$ emissions, with an $R^2$ value of up to 0.87 for shrub fires. These results provide stronger empirical support for interpreting emission trends. We also expanded the discussion on the impacts of policies on wildfire emissions. We believe the revised manuscript offers a more coherent and reliable reference for understanding wildfire $CO_2$ dynamics in China and their implications for climate policy and fire emission management. We greatly appreciate the reviewer's insights, which have significantly improved the quality of our manuscript.

**Major Comments**:

As the authors point out, there exists substantial uncertainty in wildfire emissions in existing emission inventories. In the present study, there are no attempts to quantify or report the emission

uncertainties. For instance, MODIS products usually miss the small fires (typical for croplands) due to spatial and temporal resolutions, and the classification algorithm can misclassify burnt areas/vegetation types. Without the use of additional information (e.g. ground-truth data) to calibrate and validate the burnt area product/classification algorithm, the estimated emissions raise concerns, and the robustness of the results cannot be examined. I find this to be a serious limitation of the study. As an example, how do those results vary if we use other satellite products (e.g. burnt area from FIRECCI, other NDVI) or ancillary data? What are the associated errors when using derived/literature-based combusting efficiency or emission factors? Combusting efficiency can be influenced by a number of factors, such as the intensity and type of fires, moisture content and load of combustibles, and meteorological conditions, as noted. In that case, how are those estimations impacted when eq. 2 considers only the fraction of vegetation in the empirical relationships? These issues have to be addressed before the manuscript is considered for publication.

Further, the novelty of this study and the value-addition to the existing emission estimate approaches need to be clarified. There are different global fire emission inventories available (e.g. GFED, GFAS). Also, the authors cite a set of national emission inventories in China or the sub-national level. How do the present estimations compare to those? It is unclear how the present approach is advantageous (or even different) from already existing inventory-based methods for estimating fire emissions. Moreover, the current version of the manuscript lacks a comprehensive and comparative assessment of available inventory data, especially when reporting annual/monthly emissions and their trends, as well as relating them to environmental or socioeconomic factors.

Sect.3.4 raises some concerns about reported correlations (detailed below), which need to be addressed/clarified.

Additionally, I have a concern over the title and the use of the term: 'Global carbon emission accounting'. I do believe that the study is more towards the bottom-up based wildfire emissions from China rather than describing it in the context of global carbon emissions - although it can be linked further as a value addition. Hence, I suggest the authors reconsider the title to represent the present content of the manuscript appropriately.

**Response:** We sincerely thank the reviewer for the detailed and insightful comments, which have greatly helped improve the scientific rigor and presentation clarity of this study. We fully recognize the challenges of quantifying wildfire $CO_2$ emissions and agree that incorporating uncertainty analysis, product comparisons, and validation is crucial for ensuring the robustness of bottom-up emission inventories. To this end, we have made substantial revisions to the manuscript to address each major issue raised.

Regarding the limitations of MODIS MCD64A1, we acknowledge that the MODIS MCD64A1 product has limitations in detecting small-scale or low-intensity fires, particularly in agricultural areas. To address this known limitation, we corrected the MODIS burned area using a correction factor ($\alpha_i$). The details are as follows (Lines: 123-133):

2.2.1 Burned areas

[revised manuscript text omitted]

In addition, in the revised manuscript, we conducted a comprehensive evaluation of existing inventory data, which is detailed in Section 3.6 "Comparison with other studies" (Lines: 514-538). The details are as follows:

3.6 Comparison with other studies

We compared the wildfire $CO_2$ emissions estimates in this study with several global biomass burning inventories, including the Fire Inventory from NCAR (FINN2.5, FINN_VIIRS2.5) (Wiedinmyer et al., 2023); GFED5 (van der Werf et al., 2017), the Global Fire Assimilation System (GFAS version 1.2) (Kaiser et al., 2012), and the Quick Fire Emissions Dataset (QFED version 2.5) (Koster and Darmenov, 2015), as shown in Figure 13. While all inventories exhibited consistent interannual variability, the total emission magnitudes varied substantially. Our estimates were systematically lower than those from most global datasets and were closest to GFASv1.2. FINN2.5 reported the highest values among all inventories, likely due to its use of larger burned area inputs. By incorporating VIIRS active fire detections, FINN_VIIRS2.5 showed approximately 25% higher emissions than FINN2.5. We applied GFED5 burned area data to adjust MCD64A1 burned area estimates; however, our emissions remain lower than those from GFED, which may be attributed to differences in biomass assumptions. Biomass input remains a dominant source of uncertainty in fire emissions estimates. QFED2.5 adopted a top-down approach based on fire radiative energy (FRE), and typically yields higher emission estimates (Wiedinmyer et al., 2023; Yin et al., 2019).

We further compared our estimates with other studies in China (Table 5). For forest, shrub, and grassland fires, our $CO_2$ estimates were comparable to those reported by Zhou et al. (2017) and Li et al. (2024a), slightly higher than those of Jin et al. (2022), and lower than Yin et al. (2019). The lower values in Jin et al. (2022) may result from exclusive reliance on MODIS burned area data, whereas the higher values in Yin et al. (2019) stem from the use of the FRE-based method. Regarding cropland fires, remote sensing often fails to detect small-scale agricultural burning. Consequently, many studies have used statistical data to estimate emissions, based on assumed field residue burning percentages ranging from 10% to 80% (Gao et al., 2002; Huang et al., 2012; Li et al., 2024b; Wang and Zhao, 2008; Yan et al., 2006; Yang et al., 2008; Zhou et al., 2017). The cropland fire $CO_2$ emissions estimated by the method based on burning proportion are generally higher than those calculated by the satellite remote sensing monitoring method adopted in this study.

[Figure]

Figure 13: Comparison of wildfires $CO_2$ emissions in China from multiple global inventories (2001-2022).

Table 5 Comparison of wildfires $CO_2$ emissions estimates in China from previous studies(Tg).

| Reference | year | region | Forest | Shrubland | Grassland | Cropland | All types |
|---|---|---|---|---|---|---|---|
| Wang and Zhao. (2008) | 2006 | China | - | - | - | 154.5 | - |
| Huang et al. (2012) | 2006 | China | - | - | - | 68 | - |
| This study | 2006 | China | - | - | - | 26.3 | - |
| Zhou et al. (2017) | 2012 | China | | 10.1 | 0.7 | 207.3 | - |
| This study | 2012 | China | | 8.0 | 2.4 | 35.9 | - |
| Yin et al.(2019) | 2003-2017 (mean) | China | 40.8 | 1.2 | 14.1 | 35.3 | 91.4 |
| This study | 2003-2017 (mean) | China | 23.1 | 4.0 | 0.9 | 21.7 | 49.7 |
| Jin et al.(2022) | 2001-2019 (mean) | China | | 15.2 | | - | - |
| This study | 2001-2019 (mean) | China | | 22.2 | | - | - |
| Li et al.(2024a) | 2001-2020 (mean) | Heilongjiang | 3.9 | - | 0.07 | 13.2 | - |
| This study | 2001-2020 (mean) | Heilongjiang | 4.7 | 0.3 | 0.3 | 5.0 | - |

We agree that the original version lacked sufficient discussion of the mechanisms in wildfire $CO_2$ emissions in China. In the revised manuscript, we have strengthened this analysis. These additions are now discussed in Section 3.4 of the revised manuscript. The details are as follows (Lines: 334-462):

[revised manuscript text omitted]

Note: * An asterisk next to a number indicates a statistically significant $p$-value ($p < 0.01$).

[Figure]

Figure 8: Temporal evolution of GTWR regression coefficients for wildfires $CO_2$ emissions across four vegetation cover types in China (2001–2019). Positive and negative values indicate the direction and magnitude of each variable's influence on $CO_2$ emissions.

[Figure]

Figure 9: Spatial distribution of GTWR regression coefficients for forest fire $CO_2$ emissions and their driving factors across China. The maps illustrate GTWR coefficients of six environmental and socioeconomic variables: (a) temperature, (b) relative humidity, (c) wind speed, (d) daily cumulative sunshine hours, (e) gross domestic product (GDP), and (f) population density. Grey regions represent areas where the intercept was zero (i.e., no valid model fit), and black × symbols mark locations where the regression coefficients did not pass the significance test (p ≥

0.05). Figure (g) and (h) show the model residuals and predicted shrubland $CO_2$ emissions.

[Figure]

Figure 10: Spatial distribution of GTWR regression coefficients for shrub fire $CO_2$ emissions and their driving factors across China. The maps illustrate GTWR coefficients of seven environmental and socioeconomic variables: (a) temperature, (b) precipitation, (c) relative humidity, (d) wind speed, (e) daily cumulative sunshine hours, (f) gross domestic product (GDP), and (g) population density. Grey regions represent areas where the intercept was zero (i.e., no valid model fit), and black × symbols mark locations where the regression coefficients did not pass the significance test ($p \geq 0.05$). Figure (h) and (i) show the model residuals and predicted shrubland $CO_2$ emissions.

[Figure]

Figure 11: Spatial distribution of GTWR regression coefficients for cropland fire $CO_2$ emissions and their driving factors across China. The maps illustrate GTWR coefficients of six environmental and socioeconomic variables: (a) temperature, (b) relative humidity, (c) wind speed, (d) daily cumulative sunshine hours, (e) gross domestic product (GDP), and (f) population density. Grey regions represent areas where the intercept was zero (i.e., no valid model fit), and black × symbols mark locations where the regression coefficients did not pass the significance test ($p \geq$ 0.05). Figure (g) and (h) show the model residuals and predicted shrubland $CO_2$ emissions.

[Figure]

Figure 12: Temporal trends in annual $CO_2$ emissions from cropland burning and forest, shrub, and grassland fires in China (2001–2022), and key national policy milestones related to fire and air pollution control.

**Regarding title revision**: Thank you to the reviewer for the concerns about the phrase "global carbon accounting" in the title. To better reflect the manuscript's scope, we revised the title to:

**Spatiotemporal patterns and drivers of wildfire $CO_2$ emissions in China from 2001 to 2022**
This more accurately reflects the study's focus.

We hope these revisions comprehensively address the reviewer's concerns and significantly enhance the manuscript's value in advancing understanding of wildfire $CO_2$ emissions in China.

**Comments Specific to Introduction:**

An overview of previous/available methods for estimating wildfire emissions and the comparison (or shortcomings) of those methods can be included in the introduction section.

**Response:** Thank you for your valuable suggestion. In the revised manuscript, we have enriched the Introduction section by incorporating a concise overview of existing wildfire emission estimation methods and their associated uncertainties, we have clarified in Lines 74-93:

"Traditionally, wildfire emission inventories using population or cropland area weights to allocate total emissions to grid cells have high uncertainties (Streets et al., 2003; Zhang et al., 2013). With the advancement of remote sensing technology, recent studies have shifted to satellite-based estimation methods, using active fire detection or burned area datasets to improve spatial accuracy. Inventories such as GFED (Chen et al., 2023) and the NCAR Fire Inventory (FINN) (Wiedinmyer et al., 2011) rely on satellite-derived fire count data (e.g., active fire product MCD14 ML) or burned area products (e.g., MCD64A1) to infer the timing and location of fire emissions (Giglio et al., 2016,

2018). Although satellite remote sensing has greatly improved the spatial and temporal resolution of fire detection, several practical challenges remain. For example, cloud cover, satellite overpass intervals, fire intensity thresholds, and pixel resolution can result in the underdetection of short-duration or low-intensity fires. To mitigate these limitations, this study integrated multi-source satellite products to enhance the completeness of the fire signal. Additionally, many existing global inventories rely on globally aggregated vegetation data (such as global land cover, and biomass), which further introduces errors, especially in transition zones between cropland and natural vegetation (e.g., forest-agricultural mosaics), where misclassification may lead to overestimation or underestimation of fire emissions.

To overcome these shortcomings, this study integrated China's regionally validated vegetation cover datasets (Xu et al., 2018) multi-source burned area satellite products, and regionally derived biomass data (Hu et al., 2006; Su et al., 2016; Yin et al., 2023) to develop a 500-meter resolution wildfire $CO_2$ emission inventory for China (2001-2022). Additionally, we used spatially weighted regression models to explore the drivers of emission variability and analyzed the impacts of national fire management policies on $CO_2$ emissions. The findings provide insights into the role of governance in shaping fire emissions and offer useful references for future wildfire management strategies. This multi-year emission inventory can also be used in atmospheric transport models to support the development of effective global warming mitigation strategies."

L.37: "The global annual $CO_2$ emissions from wildfires are approximately 6.5 to 11 billion tons" - please check the range (max. value); it seems to be quite large as per current records.

**Response:** Thank you for pointing out this important issue. We have verified the current estimates of global wildfire-related $CO_2$ emissions and agree that the previously cited upper limit (11 Gt $CO_2$) based on van der Werf et al. (2010) may no longer reflect the latest assessments. In the revised manuscript (Lines: 33-35), we have replaced this statement with the latest data compiled by the Global Wildfire Information System and reported by Our World in Data (2025). This correction ensures that our discussion reflects the most up-to-date understanding of global wildfire emissions. The details are as follows (Lines: 33-35):

"According to Global Wildfire Information System data compiled by Our World in Data (Our World in Data, 2025), global wildfire $CO_2$ emissions have increased since 2020, fluctuating between 5 and 7 Gt $CO_2$ per year (1 Gt = $10^{15}$ g), with record-high levels observed in 2021 and 2023."

L.62-64: However, $CO_2$ emissions from wildfires are not included in international assessments.." Doesn't seem to be true. Please rewrite or justify.

**Response:** Thank you for this insightful comment. We agree that the original phrasing may have been overly broad and potentially misleading. In the revised manuscript, we have clarified this point by specifying the context under which wildfire $CO_2$ emissions may be excluded. The new sentence now reads (Lines: 57-61):

"Despite the growing importance of wildfire $CO_2$ emissions in climate change, such emissions are often excluded from international climate frameworks, including national inventories under the United Nations Framework Convention on Climate Change (UNFCCC), due to their classification as "natural disturbances" in the Intergovernmental Panel on Climate Change (IPCC) guidelines for Land-Use Change and Forestry (LULUCF) (IPCC, 2019)."

**Comments Specific to Data + Methods:**

Currently, Sections 2.2 and 2.3 are disjoined, providing cluttered details of the approach utilized. I suggest combining the Data and Methods sections, which helps the readers to follow the methodology easily.

**Response:** Thank you for your suggestion. In the revised manuscript, we have restructured Section 2 by integrating the "Data" and "Methods" into a unified section, as recommended. This improves the logical flow and helps readers better understand the research framework and methodology. The details are as follows (Lines: 113-180):

[revised manuscript text omitted]

Also, the resolution of the final product (Ei or Ej) is not clear. From eq. 3, it might be emissions per vegetation cover or per crop type. In that case, how do you address differences in spatial resolutions, given that the vegetation cover is at 250 m, the burnt area is at 500 m, and the harvesting area dataset is at 1 km? Please clarify.

**Response:** Thank you for your question. The final product (Ei or Ej) is generated at a spatial resolution of 500 m × 500 m. All input datasets were resampled to a common resolution of 500 m × 500 m before calculation. We have clarified in Lines 121-122:

"All datasets were resampled to 500 m resolution to ensure spatial consistency."

What is Tc in eq. 2?

**Response:** Thank you for pointing this out. The term Tc in Equation 2 refers to vegetation cover fraction. It was a typographical error, and we have removed it in the revised methods section.

**Comments Specific to Results and Discussion:**

L.155: "The total $CO_2$ emissions from wildfires in China from 2001 to 2022 were 693.7 Tg, with an average annual value of 31.5 Tg, accounting for 0.46% of the total global emission of All fire types (GFED4)..." I assume that 693.7 Tg is the estimate for this study, and the percentage ( 0.46%) is calculated by considering the total global emission from GFED. How do your estimates compare with existing inventories? What is the contribution percentage of Chinese wildfire emissions according to GFED only?

**Response:** Thank you for your valuable comments. In the revised manuscript, we have added a detailed comparison between the estimates of this study and the GFED5 global fire emission inventory, as well as other China-related studies. We have added a relevant section (3.6 Comparison with other studies) to the manuscript to include this comparison and discussion (Lines: 514-538). We have also provided the detailed content of Section 3.6 "Comparison with other studies" in the comments [Major Comments].

L.157: Does 0.52% mean net forest fire emissions to fossil fuel emissions? Biomes have sequestration potential, and vegetation regrowth in later years can help offset the impact of wildfires. So, it is not straightforward to compare annual biomass emissions (only) with fossil fuel emissions to assess the biomass contribution to $CO_2$ emissions. How does the number or extent of forest fires compare with the forest regrowth patterns across China?   Please clarify.

**Response:** Thank you for your comment. We would like to clarify that the originally mentioned "0.52%" referred to the ratio of annual $CO_2$ emissions from wildfires to China's fossil fuel $CO_2$ emissions in the same year. This figure represented gross emissions, and did not account for carbon sequestration from post-fire vegetation regrowth. Therefore, it should not be interpreted as net emissions. In this study, we adopted a bottom-up approach to estimate the instantaneous release of $CO_2$. The goal was to quantify the direct contribution of wildfires to atmospheric $CO_2$ in a given year. This estimation method is consistent with global fire emission inventories such as GFED and FINN, which also report gross instantaneous emissions—that is, the total amount of carbon released to the atmosphere at the time of burning, without considering subsequent carbon uptake by

ecosystems (e.g., forest regrowth or soil carbon sequestration). We also agree with the reviewer that the long-term carbon impact of wildfires depends not only on the amount of carbon released but also on the ecosystem's recovery capacity, which varies with vegetation type, climate conditions, and post-fire regeneration. Therefore, to avoid potential misinterpretation of wildfires as a net carbon source, we have removed this comparison from the revised manuscript.

L.159: "accounting for 46% and 32%". Please correct it to: accounting for 46% and 32% of the total wildfire emissions in China."

**Response:** Thank you for your suggestion. The wording in the revised manuscript has been modified (Lines: 234-236):

"$CO_2$ emissions from cropland and forest fires were relatively high, accounting for 45% and 46% of the total wildfire emissions in China, respectively; shrub fires emissions account for 8% of the total wildfire emissions in China, while grassland fire emissions were the lowest, accounting for only 2% of the total wildfire emissions in China (Fig. 2b)."

Fig. 2: Forest fires show a declining trend, but I am not sure how this can be attributed solely to effective forestry management strategies. How about the delays in forest regrowth after the extreme fires? For example, the extreme forest fires occurred in 2003 (around Heilongjiang and Inner Mongolia Provinces) and 2008 (Inner Mongolia), as per Fig. 8. Did this study consider the effects of forest regrowth time?

**Response:** Thank you for your thoughtful question. In this study, we focused on estimating gross wildfire-related $CO_2$ emissions based on satellite-derived burned area and biomass data, and we did not explicitly model post-fire vegetation recovery or regrowth processes. Therefore, the potential delays in forest regeneration following extreme fire events in 2003 and 2008 (e.g., in Heilongjiang and Inner Mongolia) were not considered in the emission estimates. Regarding the observed decline in forest fire emissions, we agree that this trend likely reflects a combination of factors. While improvements in forest fire prevention and control policies in China may have played an important role, we also recognize that climatic and ecological factors are influential. To better understand these dynamics, we conducted spatiotemporal geographically weighted regression (GTWR) analyses incorporating climatic variables such as temperature, precipitation, humidity, wind speed, and sunshine duration. The results indicated that climate variables had significant and regionally heterogeneous impacts on wildfire $CO_2$ emissions, suggesting that the observed trends are indeed driven by multiple interacting factors—including, but not limited to, forest policy interventions. In the revised manuscript, the discussion on the influencing factors of wildfire $CO_2$ emissions in China is presented in Section 3.4 "The impact of factors on wildfires in China" (Lines: 333-462). We have also provided the detailed content of Section 3.4 "The impact of factors on wildfires in China" in the comments [Major Comments].

L.183: ".. policy management." What about the influence of other environmental conditions, such as crop types, harvesting cycles, crop-specific-suitable temperature, precipitation, water availability, etc?

**Response:** Thank you for your valuable comment. We agree that the occurrence of cropland fires is not only related to policy management but is also influenced by crop types and harvest cycles (such as temperature, precipitation, and water resources). Due to limitations in the availability of crop classification data at the national scale and over long time series, this study did not explicitly distinguish between them. However, we recognize that differences in straw yield, flammability, and

burning timing among different crops may indeed affect the spatial and temporal patterns of emissions. In the initial stage of this study, we used Pearson correlation coefficient analysis to explore the relationship between climatic factors and $CO_2$ emissions, but no significant correlation was found at the national scale, leading us to initially emphasize the role of policy management in our discussions. However, in subsequent work, we realized that this approach failed to account for the spatial and temporal heterogeneity of climate-fire relationships. Therefore, in the revised manuscript, we introduced a Geographically and Temporally Weighted Regression (GTWR) model to more appropriately characterize the dynamic relationships between climatic factors and fire emissions. In the revised manuscript, the discussion on the influencing factors of wildfire $CO_2$ emissions in China is presented in Section 3.4 "The impact of factors on wildfires in China" (Lines: 333-462). We have also provided the detailed content of Section 3.4 "The impact of factors on wildfires in China" in the comments [Major Comments].

Table 1: Not clear what additional information is conveyed through spatial autocorrelations. Please clarify.

L211: The hotspot analysis method is not clear, though the results are nicely outlined. Please include additional details on the method.

**Response:** Thank you for valuable comment. We performed global spatial autocorrelation analysis (Moran's I) to assess whether the distribution of wildfire-related $CO_2$ emissions exhibits significant spatial clustering. This step serves as a preliminary diagnostic to justify the use of local cluster detection methods such as hotspot analysis. If no significant spatial autocorrelation were observed, subsequent hotspot analysis (e.g., Getis-Ord Gi*) might not be statistically meaningful. Our results showed significant positive Moran's I values (e.g., I > 0, p < 0.01), indicating that wildfire emissions are not randomly distributed in space but tend to form clusters. This finding provides the statistical rationale for conducting the hotspot analysis presented later in the manuscript, which aims to locate areas of persistently high or low fire-related $CO_2$ emissions. We have provided a detailed description of the Moran's I index, hotspot analysis method (Getis-Ord Gi*) and GTWR in Section 2.3 (Lines:181-229). The details are as follows:

2.3 Spatiotemporal analysis of wildfire $CO_2$ emissions

2.3.1 Global spatial autocorrelation analysis

Global spatial autocorrelation is a fundamental concept in spatial statistics, used to assess the overall spatial dependence of a variable across a study region. Anselin's Moran's I index (Anselin, 1995; Moran, 1948) and the Getis-Ord Gi coefficient* (Getis and Ord, 1992) are commonly used to measure the degree of spatial clustering and heterogeneity. Moran's I is a global spatial autocorrelation statistic that quantifies the degree to which similar attribute values are clustered or dispersed in space. The Moran's I is calculated as follows:

$$I = \frac{n}{S_0} \times \frac{\sum_{i=1}^{n}\sum_{j=1}^{n} w_{ij}(x_i-\bar{x})(x_j-\bar{x})}{\sum_{i=1}^{n}(x_i-\bar{x})^2} \tag{6}$$

where I is the Global Moran's I index, n is the total number of spatial elements; $x_i$ and $x_j$ are the observed values at spatial units $i$ and $j$, respectively; $\bar{x}$ is the mean of all observed values; $w_{ij}$ is the weight matrix for the adjacency relationships between geographical units; $S_0$ is the sum of all spatial weights. The Moran's I is between -1 and 1. A value of I > 0 indicates positive spatial autocorrelation, i.e., similar values (high or low) tend to occur near each other, while I < 0 indicates dissimilar values are adjacent. I ≈ 0 suggests a random spatial pattern. Statistical significance is assessed by comparing the observed Moran's I to a null distribution generated via random

permutations. A $z$-score > 2.58 and $p$-value < 0.01 indicates a statistically significant spatial clustering pattern at the 99% confidence level. In the context of this study, significantly positive Moran's I values indicate that wildfire $CO_2$ emissions are spatially clustered, meaning that regions with high emissions tend to be adjacent to other high-emission areas, and low-emission regions are likewise grouped. This justifies further localized analyses such as hotspot detection.

2.3.2 Hot spot analysis

While Moran's I provides a global measure of spatial autocorrelation, it does not explicitly identify localized clusters of high or low values. To address this limitation, the Getis-Ord Gi* statistic(Getis and Ord, 1992) is commonly used to identify statistically significant hotspots and coldspots within spatial datasets. Unlike Moran's I, which captures both positive and negative spatial autocorrelation, the Gi* statistic focuses on detecting concentration patterns of high or low values within the study area. The Getis-Ord Gi* statistic is defined as:

$$G_i^* = \frac{\sum_j w_{ij} x_j - \bar{x} \sum_j w_{ij}}{S \sqrt{\left[\frac{n \sum_j w_{ij}^2 - \left(\sum_j w_{ij}\right)^2}{n-1}\right]}} \tag{7}$$

Where $G_i^*$ is the Getis-Ord Gi* statistic for location $i$; $x_j$ is the observed value at location $j$ (e.g., CO2 emissions); $\bar{x}$ is the global mean of the observed variable; $w_{ij}$ is the spatial weight matrix, representing the spatial relationship between locations $i$ and $j$; $n$ is the total number of spatial units; $S$ is the standard deviation of the observed values.

The Gi* statistic is essentially a ratio that compares the local sum of a variable within a specified distance to the global sum, adjusted for the number of spatial units and their spatial relationships. High positive Gi* values indicate clusters of high values (hotspots), while low negative Gi* values indicate clusters of low values (coldspots). Locations with Gi* values near zero indicate random spatial patterns without significant clustering. Statistical significance is assessed using $Z$-scores and corresponding p-values. In this study, Gi* analysis was used to detect persistent high- and low-emission clusters of wildfire $CO_2$ emissions across China from 2001 to 2022. The results provided spatially explicit insights into emission patterns and their temporal dynamics.

2.3.3 Geographically and temporally weighted regression model

To capture the spatial and temporal variations of the drivers of wildfire $CO_2$ emissions, the Geographically and Temporally Weighted Regression (GTWR) model was used (Huang et al., 2010). Unlike traditional global regression models, GTWR allows the coefficients of explanatory variables to vary across both space and time, providing a more precise estimation of the local influence of different driving factors. The GTWR model is defined as:

$$y_i = \beta_0(u_i, v_i, t_i) + \sum_k \beta_k(u_i, v_i, t_i) x_{ik} + \epsilon_i \tag{8}$$

Where $x_i$ is the response variable (wildfire CO2 emissions); $(u_i, v_i, t_i)$ are the spatial coordinates and timestamp for location $i$; $\beta_0$ is the intercept term; $\beta_k$ is the local coefficient for the $k$th explanatory variable; $x_{ik}$ is the $k$th explanatory variable; $\epsilon_i$ is the error term. The accuracy of the GTWR model depends significantly on the choice of bandwidth and kernel function, which control the spatial and temporal influence of neighboring observations. In this study, an adaptive bandwidth was used to ensure that each observation has a sufficient number of neighbors, while a tricube kernel was selected for its smooth distance decay function. The optimal bandwidth was determined using the corrected Akaike Information Criterion (AICc), a widely used criterion for model selection that balances model complexity and goodness of fit (Hurvich et al., 1998; Hurvich and Tsai, 1989). This approach enabled us to explore how the effects of climatic and

socioeconomic variables on wildfire emissions vary across regions and over time.

L245: Please rewrite clearly - there is no downward trend (in Fig. 8d and Fig.3) for Heilongjiang and Inner Mongolia regions after 2012 (after the implementation of China's strict ban on open-air biomass burning) - is there any reason for this?

**Response:** Thank you for your insightful comment. We have further discussed this in the revised manuscript (The impact of factors on wildfires in China Lines: 447-463). The details are as follows:

"Notably, northeastern China is the only region where cropland burning has increased in recent years, highlighting the need for adaptive rather than restrictive policies. As one of China's major grain-producing regions, Northeast China generates large volumes of straw. Harsh winters and short windows for straw return or removal, combined with long-established farming practices, have made complete bans on straw burning particularly challenging. Prior to strict open-burning prohibitions, farmers often burned straw in a dispersed, low-intensity manner, making detection by satellite-based fire products difficult, potentially resulting in systematic underestimation of early emissions. After the implementation of strict bans, facing growing pressure from unprocessed straw accumulation, therefore, some local governments adopted more adaptive fire management policies, such as designating burning windows under favorable meteorological conditions. These "limited and concentrated burning periods" led to spatiotemporally clustered fire events that were more easily captured by remote sensing. In recent years, the Chinese government has also promoted the scientific incorporation of straw into soils, off-field collection, and the industrial utilization of crop residues in Northeast China. These efforts highlight the significant role of policy in shaping emission trends from agricultural burning, particularly in regions where environmental constraints and traditional farming practices pose unique challenges."

Sect. 3.4: The use of vegetarian primary productivity and the NDVI in factor analyses is inappropriate. They are already part of the wildfire emission calculation; hence, they must be highly correlated with emissions. Please clarify.

**Response:** Thank you for pointing this out. We agree with the reviewer's concern that including vegetation productivity (such as NDVI and NPP) in the impact factor analysis is inappropriate, as these variables are directly or indirectly involved in the estimation of wildfire $CO_2$ emissions. In the revised manuscript, we have removed the Normalized Difference Vegetation Index (NDVI) and Net Primary Productivity (NPP) from the set of independent variables used in the regression and spatial analyses to avoid redundancy and potential collinearity issues. This revision enhances the robustness and interpretability of the impact factor analysis. In the revised manuscript, the discussion on the influencing factors of wildfire $CO_2$ emissions in China is presented in Section 3.4 "The impact of factors on wildfires in China" (Lines: 333-462). We have also provided the detailed content of Section 3.4 "The impact of factors on wildfires in China" in the comments [Major Comments].

Sect. 3.4: "… daily cumulative sunshine hours (forest:-0.41, shrub:0.25; p < 0.001) …, while the main factor affecting $CO_2$ emissions from grassland fires was temperature (-0.45, p < 0.001)". How have these correlation coefficients become negative?

**Response:** Thank you for your thoughtful question. These coefficients were initially obtained through Spearman's rank correlation, a method that assumes the independence of observations. However, due to the presence of spatial autocorrelation in our data, this assumption is not valid, potentially leading to inflated significance levels and misleading correlation signs. To address this issue and better account for spatiotemporal heterogeneity, we have replaced the correlation analysis

with a Geographically and Temporally Weighted Regression (GTWR) model in the revised manuscript. In the revised manuscript, the discussion on the influencing factors of wildfire $CO_2$ emissions in China is presented in Section 3.4 "The impact of factors on wildfires in China" (Lines: 333-462). We have also provided the detailed content of Section 3.4 "The impact of factors on wildfires in China" in the comments [Major Comments].

L275: "An increase in GDP and population density was often accompanied by better agricultural technology and management practices, including more effective management alternatives to straw burning." Please backup.

**Response:** Thank you for your insightful comment. We agree that the original description of the relationship between GDP/population density and improved agricultural practices required more robust justification. We reviewed several studies on straw burning in China and found that more economically developed regions in China do tend to adopt alternative straw management practices such as straw returning to fields or off-site utilization. For example, a study in Hunan Province found a negative correlation between GDP, population density, and the incidence of straw burning. In the revised manuscript, we re-conducted the impact factor analysis of wildfire emissions using geographically and temporally weighted regression (GTWR). In the revised manuscript, the discussion on the influencing factors of wildfire $CO_2$ emissions in China is presented in Section 3.4 "The impact of factors on wildfires in China" (Lines: 333-462). We have also provided the detailed content of Section 3.4 "The impact of factors on wildfires in China" in the comments [Major Comments].

L280: "China's policies have also significantly reduced $CO_2$ emissions from opening biomass burning fires." This is quite contradictory to Fig. 3

**Response:** Thank you for pointing out this important inconsistency. We agree that the original statement "China's policies have significantly reduced $CO_2$ emissions from open biomass burning" was overly general and did not fully reflect the regional disparities. In the revised manuscript, we have rephrased this statement to more accurately reflect the spatial heterogeneity in emission trends. In the revised manuscript, the discussion on the influencing factors of wildfire $CO_2$ emissions in China is presented in Section 3.4 "The impact of factors on wildfires in China" (Lines: 433-462). We have also provided the detailed content of Section 3.4 "The impact of factors on wildfires in China" in the comments [Major Comments].

L311: "However, the current international assessment and national emission reduction responsibilities do not include wildfire carbon emissions or consider measures such as reducing wildfire frequency and intensity through wildfire management." I'm not sure if this is entirely correct.

**Response:** Thank you for this important observation. We agree that our original statement may have been overly absolute. While it is true that wildfire emissions are not consistently included in formal national greenhouse gas (GHG) inventories under the UNFCCC, some countries do report wildfire-related $CO_2$ emissions in the Land Use, Land-Use Change and Forestry (LULUCF) sector. However, these are often treated as natural disturbances and excluded from anthropogenic emission reduction targets, unless directly influenced by human activity or managed interventions. In the revised manuscript, we have modified this statement to reflect this nuance more accurately.

**Comments Specific to Conclusion:**

L329: "Human activities significantly influence CO2 emissions from cropland fires. Emissions negatively correlated with GDP (-0.52) and population density (-0.51)." These two sentences seem

to be contradictory.

L330: "Various factors, such as accumulated sunshine hours (-0.41, p < 0.001) …), mainly influenced emissions from forest and shrub fires, while temperature (-0.45, p < 0.001) primarily affected emissions from grassland fires." Please see my comments above - why are they negatively correlated?

**Response:** Thank you for your follow-up comment regarding the direction of the correlation. As mentioned in our previous response, the original correlation values were obtained through Spearman's correlation analysis, which assumes the independence of observations. However, due to the presence of spatial autocorrelation in the emission and climate data, this assumption may not hold, potentially leading to biases in the direction of the correlation or an overestimation of significance levels. In the revised manuscript, we have replaced this method with a Geographically and Temporally Weighted Regression (GTWR) model, which accounts for spatial heterogeneity and allows for a more accurate estimation of the impact of various factors.

**Reviewer #2:**

The study presents a well-structured and timely study on wildfire $CO_2$ emissions in China, using satellite data and a bottom-up approach to analyze spatiotemporal trends from 2001 to 2022. The topic is highly relevant to climate change mitigation and environmental governance, and the methodology appears robust. However, several aspects could be clarified or expanded to strengthen the manuscript's impact and scientific rigor.

(1) The authors should reconsider the title of the manuscript. While this study focuses exclusively on wildfire $CO_2$ emissions in China, the current title, "Global carbon emission accounting: national-level assessment of wildfire $CO_2$ emission—a case study of China", suggests a broader global scope. This could be misleading to readers, as the study does not provide data or analysis on global carbon emissions or wildfire $CO_2$ emissions beyond China. If this is part of a series of publications on global carbon emission accounting with a focus on individual countries, it would be helpful to clarify this in the title. A more precise and region-specific title would better reflect the study's content and avoid confusion.

**Response:** We greatly appreciate your insightful suggestions regarding the manuscript title. We fully agree with the reviewer's perspective that the original title might mislead readers. To address this issue and more accurately reflect the content and scope of the paper, we have revised the title to:

"Spatiotemporal patterns and drivers of wildfire $CO_2$ emissions in China from 2001 to 2022"

We are extremely grateful for the reviewer's recommendations, which have helped enhance the clarity and accuracy of the manuscript's presentation.

(2) This study only presents the average $CO_2$ emissions from wildfires but does not estimate the uncertainties propagating from emission factors and activity data. Without such an uncertainty assessment, readers cannot evaluate the reliability and accuracy of the wildfire $CO_2$ emission estimates. Consequently, the claim on page 18, lines 306–307, that "wildfire $CO_2$ emissions provide accurate input data for simulating the effects of wildfires on air quality, climate, and human health" warrants reconsideration. The reviewer suggests to include an uncertainty assessment, which would significantly strengthen the study's conclusions and enhance its credibility for use in simulation models. In addition, the uncertainty of $CO_2$ emissions can be included in figures, for example, Figure 2, Figure 3.

**Response:** Thank you for your valuable suggestion. In the revised manuscript, we have conducted a Monte Carlo uncertainty analysis to quantify the uncertainty associated with emission factors and activity data. The results have been incorporated into a new section 3.5 "Uncertainty analysis" (Lines: 463-513). Additionally, to better communicate the uncertainties in our estimates, we have revised the figures accordingly. Figure 2 and Figure 3 have been merged into Figure 2, and we now include a visual representation of the uncertainty range: "The grey shaded area represents the 5th-95th percentile confidence interval derived from the Monte Carlo uncertainty analysis" (Lines: 237-244). The details are as follows:

3.5 Uncertainty analysis

[revised manuscript text omitted]

Figure 2: (a) Annual $CO_2$ emissions within specific vegetation cover types from 2001 to 2022 in China; (b) Contribution of different vegetation cover types to the total CO2 emissions from 2001 to 2022 in China. (c–f) Time series of $CO_2$ emissions for forest, shrub, grassland and cropland, respectively; the red dashed line is the linear trend and the grey shaded envelope represents the 5th-95th percentile confidence interval from Monte Carlo uncertainty analysis; the p-values are derived from the Mann-Kendall trend test, a non-parametric statistical method used to assess the presence of a monotonic (increasing or decreasing) trend in a time series without assuming any specific data distribution. A p-value < 0.05 indicates a statistically significant trend at the 95% confidence level.

(3) The reviewer suggests that the authors consider reorganizing method Section 2.3 to improve the logical flow. Specifically, moving Section 2.3.4 ($CO_2$ emission estimation) to the beginning of Section 2.3, before Section 2.3.1 (Emission Factors), would create a more coherent structure. This reorganization would allow the authors to first present the overall approach for estimating $CO_2$ emissions, followed by the details of the input parameters, such as emission factors (Section 2.3.1) and activity data (Sections 2.3.2 and 2.3.3). This adjustment would enhance the clarity and readability of the Methods section, making it easier for readers to follow.

**Response:** Thank you for your constructive suggestion. In the revised manuscript, we have reorganized Section 2.2 as recommended. The description of the overall $CO_2$ emission estimation method now appears at the beginning of the section (as the new Section 2.2.1), followed by the details on emission factors and activity data. This adjustment improves the clarity and logical flow of the methods, making it easier for readers to understand the structure of the emission estimation framework. The details are as follows:

2.2 $CO_2$ emission estimation

    2.2.1 Burned areas

    2.2.2 Emission factors

    2.2.3 Aboveground biomass

    2.2.4 Combustion efficiency

(4) In the in-text citations and reference list, the surnames "van der Werf" and "van Wees" are incorrectly formatted, with "van der", "van", capitalized. According to academic citation conventions, prefixes like "van der", "van", and similar are not capitalized unless they appear at the beginning of a sentence. These prefixes are considered part of the Dutch surname but are not treated as independent proper nouns. Correcting this will ensure the citations follow established formatting standards and provide proper attribution to the authors.

**Response:** Thank you for pointing out this citation formatting issue. We have carefully reviewed and corrected all instances of Dutch surnames in both the in-text citations and reference list in accordance with academic style guidelines. The prefixes such as "van" and "van der" are now consistently formatted in lowercase, except when appearing at the beginning of a sentence.

(5) Page 3, Line 79: Replace "The study results can provide…" with "The results of this study can provide"

**Response:** Thank you for the suggestion. The sentence on line 79 has been revised for clarity.

(6) Page 3, Line87: Add "geographical" before the "distribution" in the sentence "There are differences in the distribution of cropland, grassland, shrubs, and forests in China."

**Response:** Thank you for the correction. We have inserted "geographical" before "distribution" to improve clarity (Line 99).

(7) Page 4, Line 101: Replace "spatial" with "spatiotemporal" in the sentence "Vegetation cover data were combined with fire area data to extract spatial data, including the time and geographic coordinates of fire occurrence, burned area,"

**Response:** Thank you for the suggestion. In the revised manuscript, this section has been reorganized and merged into the methods section based on suggestions from other reviewers. The original sentence has been restructured and the terminology has been updated accordingly. The revised text now more accurately reflects the integration of both spatial and temporal fire data.

(8) Page 5, Line 126: The citation for the China Statistical Yearbook should be included both as an in-text citation and in the reference list.

**Response:** Thank you for your comment. We have corrected this in the revised manuscript by adding an in-text citation for the China Statistical Yearbook and including the full reference in the reference list (Lines: 738-739).

"NBSC (National Bureau of Statistics of China): China Statistical Yearbook (2001-2022), National Bureau of Statistics of China, Beijing, China."

(9) Page 6, Line 137: "Combustion efficiency" should more appropriately be abbreviated as "CE" rather than "CF".

**Response:** Thank you for pointing this out. We have corrected the abbreviation throughout the manuscript, replacing "CF" with "CE" where referring to combustion efficiency to ensure consistency and accuracy.

(10) Page 9, Line 183–185: The manuscript states that "The high emissions of cropland fires in March and April mainly originated from Heilongjiang and Jilin Provinces. The high emissions of cropland fires in May and June mainly came from the Anhui, Henan, and Jiangsu Provinces." Could the authors explain why the cropland fire emissions peak at different months between the two regions ("Heilongjiang and Jilin Provinces" vs. "Anhui, Henan, and Jiangsu Provinces")?

**Response:** Thank you for this thoughtful question. We have added this explanation to the revised manuscript to clarify the observed seasonal differences in cropland fire emissions (Lines: 263-269):

"In contrast, the spatial distribution of emissions from cropland fires varied significantly across different months and was likely influenced by policy management (Fig.4). High emissions in March and April were concentrated in NE region, while emissions in May and June were primarily associated with the NC region. The regional difference in peak emission months can be attributed to distinct cropping systems and climatic conditions. In the NE region (e.g., Heilongjiang and Jilin), cold winters and delayed spring thaw often push straw burning activities into March-April,

following the autumn harvest. In contrast, the NC region (e.g., Anhui, Henan, Jiangsu) practices a double-cropping system of winter wheat and summer maize, where wheat is harvested in May–June, and burning of straw residues is typically observed during this transition period."

(11) Throughout the manuscript, the "p" in p-values should be italicized to align with standard formatting conventions in scientific writing.

**Response:** Thank you for pointing this out. We have corrected the formatting throughout the manuscript by italicizing all instances of "p" in p-values to align with scientific style guidelines.

(12) Page 11, Line 202: do the authors intent to say "national wide" instead of "global"? Again, the CO2 emission in China is clearly not global. In addition, the spatial autocorrelation analysis (now in the supplement) and Table 1 should be moved to the method section in the main text. Further, the authors state that "the p values were all less than 0.01, with a confidence level of 99%; the Moran's I values were all positive, with a Z score greater than 2.58, indicating a significant positive spatial autocorrelation of $CO_2$ emissions from wildfires, exhibiting an aggregation pattern in spatial distribution." However, the terms "positive spatial autocorrelation" and "aggregation pattern in spatial distribution" are not clearly explained in the manuscript. Could the authors elaborate on what these terms specifically mean in the context of wildfire $CO_2$ emissions?

**Response:** Thank you for your thoughtful comments. We would like to clarify that the term "global spatial autocorrelation" refers to global (or overall) Moran's I analysis in spatial statistics, and does not imply global geographic coverage. In this context, "global" refers to the entire study region (China), in contrast to "local" indicators such as Getis-Ord Gi*. To avoid confusion, we have revised the manuscript to clarify this terminology. Regarding the interpretation of "positive spatial autocorrelation" and "clustered spatial pattern", we have added a brief explanation to the methods (Lines: 181-229). In line with your recommendation, we have also moved the spatial autocorrelation analysis from the Supplementary Materials into the main Methods section (Section 2.3) to improve clarity and accessibility. The details are as follows:

2.3 Spatiotemporal analysis of wildfire $CO_2$ emissions

2.3.1 Global spatial autocorrelation analysis

Global spatial autocorrelation is a fundamental concept in spatial statistics, used to assess the overall spatial dependence of a variable across a study region. Anselin's Moran's I index (Anselin, 1995; Moran, 1948) and the Getis-Ord Gi coefficient* (Getis and Ord, 1992) are commonly used to measure the degree of spatial clustering and heterogeneity. Moran's I is a global spatial autocorrelation statistic that quantifies the degree to which similar attribute values are clustered or dispersed in space. The Moran's I is calculated as follows:

$$I = \frac{n}{S_0} \times \frac{\sum_{i=1}^{n}\sum_{j=1}^{n} w_{ij}(x_i - \bar{x})(x_j - \bar{x})}{\sum_{i=1}^{n}(x_i - \bar{x})^2} \tag{6}$$

where I is the Global Moran's I index, n is the total number of spatial elements; $x_i$ and $x_j$ are the observed values at spatial units $i$ and $j$, respectively; $\bar{x}$ is the mean of all observed values; $w_{ij}$ is the weight matrix for the adjacency relationships between geographical units; $S_0$ is the sum of all spatial weights. The Moran's I is between -1 and 1. A value of I > 0 indicates positive spatial autocorrelation, i.e., similar values (high or low) tend to occur near each other, while I < 0 indicates dissimilar values are adjacent. I ≈ 0 suggests a random spatial pattern. Statistical significance is assessed by comparing the observed Moran's I to a null distribution generated via random permutations. A $z$-score > 2.58 and $p$-value < 0.01 indicates a statistically significant spatial

clustering pattern at the 99% confidence level. In the context of this study, significantly positive Moran's I values indicate that wildfire $CO_2$ emissions are spatially clustered, meaning that regions with high emissions tend to be adjacent to other high-emission areas, and low-emission regions are likewise grouped. This justifies further localized analyses such as hotspot detection.

**2.3.2 Hot spot analysis**

While Moran's I provides a global measure of spatial autocorrelation, it does not explicitly identify localized clusters of high or low values. To address this limitation, the Getis-Ord Gi* statistic(Getis and Ord, 1992) is commonly used to identify statistically significant hotspots and coldspots within spatial datasets. Unlike Moran's I, which captures both positive and negative spatial autocorrelation, the Gi* statistic focuses on detecting concentration patterns of high or low values within the study area. The Getis-Ord Gi* statistic is defined as:

$$G_i^* = \frac{\sum_j w_{ij} x_j - \bar{x} \sum_j w_{ij}}{S \sqrt{\left[\frac{n \sum_j w_{ij}^2 - \left(\sum_j w_{ij}\right)^2}{n-1}\right]}} \tag{7}$$

Where $G_i^*$ is the Getis-Ord Gi* statistic for location $i$; $x_j$ is the observed value at location $j$ (e.g., CO2 emissions); $\bar{x}$ is the global mean of the observed variable; $w_{ij}$ is the spatial weight matrix, representing the spatial relationship between locations $i$ and $j$; $n$ is the total number of spatial units; $S$ is the standard deviation of the observed values.

The Gi* statistic is essentially a ratio that compares the local sum of a variable within a specified distance to the global sum, adjusted for the number of spatial units and their spatial relationships. High positive Gi* values indicate clusters of high values (hotspots), while low negative Gi* values indicate clusters of low values (coldspots). Locations with Gi* values near zero indicate random spatial patterns without significant clustering. Statistical significance is assessed using Z-scores and corresponding p-values. In this study, Gi* analysis was used to detect persistent high- and low-emission clusters of wildfire $CO_2$ emissions across China from 2001 to 2022. The results provided spatially explicit insights into emission patterns and their temporal dynamics.

**2.3.3 Geographically and temporally weighted regression model**

To capture the spatial and temporal variations in wildfire $CO_2$ emissions, the Geographically and Temporally Weighted Regression (GTWR) model was used(Huang et al., 2010). Unlike traditional global regression models, GTWR allows the coefficients of explanatory variables to vary across both space and time, providing a more precise estimation of the local influence of different driving factors. The GTWR model is defined as:

$$y_i = \beta_0(u_i, v_i, t_i) + \sum_k \beta_k(u_i, v_i, t_i) x_{ik} + \epsilon_i \tag{8}$$

Where $x_i$ is the response variable (wildfire CO2 emissions); $(u_i, v_i, t_i)$ are the spatial coordinates and timestamp for location $i$; $\beta_0$ is the intercept term; $\beta_k$ is the local coefficient for the $k$th explanatory variable; $x_{ik}$ is the $k$th explanatory variable; $\epsilon_i$ is the error term. The accuracy of the GTWR model depends significantly on the choice of bandwidth and kernel function, which control the spatial and temporal influence of neighboring observations. In this study, an adaptive bandwidth was used to ensure that each observation has a sufficient number of neighbors, while a tricube kernel was selected for its smooth distance decay function. The optimal bandwidth was determined using the corrected Akaike Information Criterion (AICc), a widely used criterion for model selection that balances model complexity and goodness of fit(Hurvich et al., 1998; Hurvich and Tsai, 1989). This approach enabled us to explore how the effects of climatic and socioeconomic variables on wildfire emissions vary across regions and over time.

(13) Page 13, Line 244-245: The manuscript states that "After the implementation of China's strict ban on open-air biomass burning in 2012, emissions decreased, showing an overall downward trend." However, the Fig. 8d shows the opposite for Inner Mongolia that the cropland CO2 emissions in Inner Mongolia increased strongly from 2012. Could the authors explain why is it?

**Response:** Thank you for your careful observation. We hereby clarify that the region showing an increasing trend in cropland $CO_2$ emissions after 2012 is Heilongjiang Province, not Inner Mongolia. This trend has been addressed in the revised manuscript. We have clarified this point in the revised text and ensured that the differences in regional trends are clearly communicated to avoid similar confusion (Lines: 447-462):

"Notably, northeastern China is the only region where cropland burning has increased in recent years, highlighting the need for adaptive rather than restrictive policies. As one of China's major grain-producing regions, Northeast China generates large volumes of straw. Harsh winters and short windows for straw return or removal, combined with long-established farming practices, have made complete bans on straw burning particularly challenging. Prior to strict open-burning prohibitions, farmers often burned straw in a dispersed, low-intensity manner, making detection by satellite-based fire products difficult, potentially resulting in systematic underestimation of early emissions. After the implementation of strict bans, facing growing pressure from unprocessed straw accumulation, therefore, some local governments adopted more adaptive fire management policies, such as designating burning windows under favorable meteorological conditions. These "limited and concentrated burning periods" led to spatiotemporally clustered fire events that were more easily captured by remote sensing. In recent years, the Chinese government has also promoted the scientific incorporation of straw into soils, off-field collection, and the industrial utilization of crop residues in Northeast China. These efforts highlight the significant role of policy in shaping emission trends from agricultural burning, particularly in regions where environmental constraints and traditional farming practices pose unique challenges."

(14) Page 14, Line 253: The statement "High emissions still existed in Heilongjiang and Inner Mongolia in the east" is somewhat unclear. It is not immediately apparent whether "in the east" refers to the eastern parts of Heilongjiang and Inner Mongolia, or whether it refers to these regions being in the eastern part of a larger geographical context (e.g., China).

**Response:** Thank you for pointing out this ambiguity. We agree that the phrase "in the east" was vague and could be misinterpreted in the context of regional emission patterns. In the revised manuscript, we have removed this sentence entirely to avoid confusion. Additionally, we have restructured the discussion by integrating the temporal trend analysis of emissions (previously discussed here) into the section on policy impacts. This change improves the clarity and coherence of the narrative by linking emission changes more directly to relevant regulatory milestones. As discussed in detail in our response to Comment (15). We believe this revision enhances both the readability and scientific rigor of the manuscript.

(15) Page 16, Line 280: The statement "China's policies have also significantly reduced $CO_2$ emissions from opening biomass burning fires" requires clarification. Are these policies specifically targeting cropland fires, or do they apply more broadly? If the focus is on cropland fires, the sentence should be revised to clarify. Additionally, it should be noted that, in principle, all wildfires are open biomass burning.

**Response:** Thank you for your insightful comments. We acknowledge that the original statement was overly broad and lacked specificity regarding policy targets. In the revised manuscript, we have incorporated this analysis into Section 3.4. The details are as follows (Lines: 438-462):

"In China, the role of policy is particularly significant. Wu et al. (2018) reviewed 51 crop straw management regulations issued between 1965 and 2015, with 34 implemented after 2008. The timing of these intensive regulatory efforts closely aligns with key turning points in emission trends (Fig. 12). For cropland fires, annual $CO_2$ emissions increased from 8.2 Tg/year during 2001-2005 to 26.2 Tg/year during 2010-2016, but began to decline following the revision of the Air Pollution Prevention and Control Law in 2015 and the launch of the Air Pollution Action Plan in 2013. Similarly, after the implementation of the National Forest Fire Prevention Plans in 2009 and 2016, $CO_2$ emissions from forest, shrub and grassland fires dropped from 38.1 Tg/year (2006–2009) to 13.3 Tg/year (2017–2022). Jin et al. (2022) further estimated that over 80% of wildfire-related $CO_2$ emissions could be avoided under effective fire management. These findings strongly indicate that policy management plays a critical role in wildfire $CO_2$ emissions.

Notably, northeastern China is the only region where cropland burning has increased in recent years, highlighting the need for adaptive rather than restrictive policies. As one of China's major grain-producing regions, Northeast China generates large volumes of straw. Harsh winters and short windows for straw return or removal, combined with long-established farming practices, have made complete bans on straw burning particularly challenging. Prior to strict open-burning prohibitions, farmers often burned straw in a dispersed, low-intensity manner, making detection by satellite-based fire products difficult, potentially resulting in systematic underestimation of early emissions. After the implementation of strict bans, facing growing pressure from unprocessed straw accumulation, therefore, some local governments adopted more adaptive fire management policies, such as designating burning windows under favorable meteorological conditions. These "limited and concentrated burning periods" led to spatiotemporally clustered fire events that were more easily captured by remote sensing. In recent years, the Chinese government has also promoted the scientific incorporation of straw into soils, off-field collection, and the industrial utilization of crop residues in Northeast China. These efforts highlight the significant role of policy in shaping emission trends from agricultural burning, particularly in regions where environmental constraints and traditional farming practices pose unique challenges."

[Figure]

Figure 12. Temporal trends in annual $CO_2$ emissions from cropland burning and forest, shrub, and grassland fires in China (2001–2022), and key national policy milestones related to fire and air pollution control.

(16) Inconsistency between text and figure. line 295-206 states "Since 2012, following the implementation of policies for air pollution prevention and control, $CO_2$ emissions from cropland fires have decreased (Fig. 7d)", however, Figure 3d appears to show the opposite trend. Additionally, Figure 7d does not show the temporal variations of $CO_2$ emissions.

**Response:** Thank you for highlighting this important inconsistency. We acknowledge that the original statement was overly general and incorrectly referenced Figure 7d instead of Figure 3d. In the revised manuscript, we have corrected the figure citation and clarified the interpretation to better reflect the data presented.

**Reviewer #3:**

**GENERAL:**

This study analyzes the spatiotemporal variation of carbon dioxide emissions from wildfires in China from 2001 to 2022 based on high-resolution satellite-derived data and emission factors of various vegetation cover types, then further examine the correlation between wildfire emissions and multiple meteorological and anthropogenic factors. Overall, this study makes a valuable contribution by providing a comprehensive, long-term dataset with high temporal and spatial resolution, which enhances our understanding of the evolution of wildfire emissions across different ecosystems in China. Compared with previous studies, the differentiation of emission contributions from forest, shrubland, grassland, and cropland can suggest finely the contribution of different vegetation types. Furthermore, the use of spatial autocorrelation analysis (Moran's I) is relatively uncommon in existing wildfire emission studies. However, a few minor issues should be addressed before the manuscript can be considered for publication in ACP.

First, the relationship between $CO_2$ emissions and air quality assessment is unclear. While the study focuses exclusively on $CO_2$ emissions, it repeatedly mentions potential implications for air quality assessment. However, this relationship is not clearly defined in the current version of the manuscript. Since $CO_2$ is not a traditional air pollutant in the context of air quality standards, the relevance of this link should either be clarified or reconsidered to avoid conceptual ambiguity.

**Response:** Thank you for this important comment. We fully agree that $CO_2$ is not classified as a conventional air pollutant in the context of air quality standards. In the original manuscript, our references to air pollution policies were intended to highlight the regulatory background and motivations behind biomass burning bans, which were primarily enacted to control $PM_{2.5}$ and other air pollutants. In the revised manuscript, we have clarified this distinction.

Second, there are some insufficient in interpretation of spatial correlation analysis. The current discussion is limited to high-value clusters. It may also be important to consider spatial outliers, such as high-emission areas surrounded by low-emission regions. These areas could represent localized fire hotspots or regions vulnerable to extreme events, and further discussion would enrich the interpretation of spatial dynamics.

**Response:** Thank you for this insightful suggestion. We agree that spatial outliers, particularly high-emission anomalies in low-emission environments, are crucial for understanding localized fire dynamics. In the revised manuscript, we have expanded the discussion to analyze wildfire events in abnormal years within high-emission regions. The details are as follows (Lines: 296-332):

"Based on the Getis-Ord Gi* analysis (Fig. 6), we identified clear spatial clusters of persistent high (hotspots) and low (coldspots) wildfire $CO_2$ emissions (Fig. 6). Among different vegetation types, $CO_2$ emissions from forest fires are mainly distributed in NE, SW, and SC regions, with the NE region accounting for 56% of China's annual average emissions from 2001 to 2022 (Fig. 5b). The NE region (e.g., the Greater and Lesser Khingan Mountains) is a typical coniferous forest belt with abundant fuel accumulation, dry and windy spring conditions, and makes it highly prone to intense wildfires (Lian et al., 2024a). However, despite the high forest fire emissions in NE, no significant hotspots were detected by the Getis-Ord Gi* analysis (Fig. 6b), indicating that its high emissions mainly stem from sporadic extreme events rather than persistent clustering (Fig. 7a). For example, in 2003 and 2008, extreme wildfires in NE China contributed 73% and 56% of the national forest fire $CO_2$ emissions in 2003 and 2008, respectively (Fig. 7a). In contrast, forest fire $CO_2$

emissions in SW and SC exhibited spatial clustering. Forest fires in these regions are prone to occurring in late winter and early spring each year, with relatively small fire scales but high frequency (Qin et al., 2014; Zhang et al., 2023a).

Shrub fire emissions were concentrated in the SW and SC regions, accounting for 47% and 27% of China's annual average emissions from 2001 to 2022, respectively (Fig. 5b). Secondary vegetation such as shrubs and bamboo forests are common in these areas, resulting from land use changes (e.g., farmland abandonment, forest degradation), which facilitates the accumulation of combustibles (Han et al., 2018). Meanwhile, complex terrain and high biomass also amplify the risk of fire spread (He et al., 2024). Additionally, seasonal drought (low humidity) combined with human activities such as fuelwood collection and traditional burning practices (Ying et al., 2021) exacerbate fire occurrences, forming persistent spatial clustering that has been clearly identified as hotspot areas (Fig. 6).

Grassland fire emissions were mainly concentrated in the NE region, accounting for 70% of the national annual mean during 2001–2022 (Fig. 5d), with hotspot areas focusing on the temperate grasslands of Inner Mongolia (e.g., Hulunbuir, Xilingol). In this region, dry herbaceous vegetation, strong winds, and low humidity in spring make grassland fires extremely prone to ignition (Chang et al., 2023). Additionally, there is a close relationship between land use and grassland fire occurrence(Li et al., 2017). Li et al.(2017) explored the relationship between land use and the spatial distribution of grassland fires, and the results showed that land use has a significant impact on grassland fires. Grassland fires in the NE are frequent and spatially concentrated, forming significant hotspots (Fig. 6d).

High $CO_2$ emissions from cropland fires were concentrated in NC and NE regions, accounting for 51% and 42% of the national annual mean emissions, respectively, during 2001–2022 (Fig. 5e). Spatiotemporally, from 2003 to 2012, the main emission sources were agricultural provinces in NC (e.g., Hebei, Shandong, Henan, Anhui), while after 2012, agricultural regions in NE (e.g., Heilongjiang, Jilin, Liaoning) became the primary sources of emissions (Fig. 7d). These areas have high crop straw yields and long-standing traditional burning practices, making them typical hotspots of agricultural fires(Li et al., 2024a; Wu et al., 2018)."

Finally, the mechanism analysis in this study is insufficient. For example, this work attributes a reduction in agricultural fire emissions to China's 2012 policy banning open-air biomass burning. While this explanation is reasonable for cropland-related fires, the simultaneous decline in emissions from other land types (forest, shrub, and grassland) around the same period is not addressed. Exploring additional factors, such as changes in land management, fire suppression efforts, or climatic influences could strengthen the mechanism analysis.

**Response:** Thank you for this important observation. We agree that the original version lacked sufficient discussion of the mechanisms behind the decline in wildfire $CO_2$ emissions from non-cropland vegetation types. In the revised manuscript, we have strengthened this analysis in two ways. These additions are now discussed in Section 3.4 of the revised manuscript. The details are as follows (Lines: 333-462):

[revised manuscript text omitted]

**SPECIFIC**

Line 81: Consider revising "air pollution control strategies" to "global warming control strategies".

**Response:** Thank you for your helpful suggestion. We agree that "global warming control strategies" more accurately reflects the focus of our study on $CO_2$ emissions and their climate relevance. We have revised the wording accordingly in the updated manuscript.

Line 275: Lower emissions in high-GDP regions may be due to a smaller proportion of cropland in these areas. To assess the impact of farmland management practices, it is suggested to analyze emissions per unit area in different area.

**Response:** Thank you for your insightful suggestion. We agree that spatial variations in cropland area may affect total $CO_2$ emissions and could potentially confuse interpretations based solely on GDP or population density. In the revised manuscript, we addressed this issue by introducing a Geographically and Temporally Weighted Regression (GTWR) model to analyze the impacts of various factors (including GDP and population density) on wildfire $CO_2$ emissions across different land cover types (forest, shrubland, grassland, and cropland). As discussed in detail in our response to Comment [Finally, the mechanism analysis in this study is insufficient. For example, this work…].

Table 2: The bottom border line of this table is missing and should be corrected.

**Response:** Thank you for your careful observation. We have revised the formatting of Table 2 in the revised manuscript to include the missing bottom border line, ensuring consistency with journal formatting standards.

**Reviewer #4:**

**General Comments**

This manuscript presents a national-scale inventory of wildfire $CO_2$ emissions in China using a bottom-up approach based on multiple observed datasets. In specific, they used MODIS burned area data, vegetation and NDVI data, land cover data, data on farms and harvesting, meteorological data and empirical emission factors. In no cases were direct observations of fires or species emitted from the fires observed, even though there is extensive literature looking at the problem from this perspective. Overall, the topic is both of interest and relevant to the larger community. The authors also did a great job gathering and applying datasets that spanned over two decades.

However, the manuscript suffers from multiple critical limitations that currently preclude publication. In specific, the study lacks methodological novelty and primarily replicates well-established techniques without introducing new approaches or refinements. Key variables—such as emission factors, biomass, and combustion efficiency—are treated as fixed, without uncertainty quantification or regional calibration. The analysis remains largely descriptive, and the use of bivariate correlations (e.g., Spearman) to explore emission drivers is insufficient, especially given the presence of spatial autocorrelation. Moreover, important limitations of the MODIS burned area product are not addressed, particularly its tendency to underestimate small fires in cropland regions, obscured fires, fires occurring under cloudy conditions or wet surfaces both of which are commonly found throughout areas in eastern and southern China. The policy implications presented are not supported by causal analysis or empirical evaluation. In addition, the integration of multi-resolution datasets lacks clarity, raising concerns about data consistency. I have seen photographs published which demonstrate fires in an area of Southern China which does not show up on your map. Uncertainties of this nature need to at least be written about and limitations discussed.

Overall, the manuscript requires substantial revisions in terms of methods, statistical analysis, and interpretation to reach the level of scientific rigor expected for publication in ACP. I am happy to look at a future revision if all of these points are addressed.

**Response:** We sincerely appreciate the reviewer's comprehensive and in-depth evaluation. Your feedback has been invaluable in helping us identify key areas for improvement in methodology, analytical depth, and interpretation. In response, we have made substantial revisions throughout the manuscript to address the main issues raised. These modifications include conducting a comprehensive Monte Carlo uncertainty analysis for critical input variables, replacing simple correlation analysis with spatial regression models (OLS, GWR, GTWR), and incorporating multi-source comparisons and corrections to address underestimation issues with MODIS burned area products. Additionally, we provide more detailed region-specific explanations for the spatial clustering of emissions, link observed trends to relevant policy interventions, and carefully evaluate the consistency and applicability of input datasets. While the core estimation framework remains based on established bottom-up approaches used in global inventories, we have introduced China-specific adaptations and refined uncertainty handling. We believe these improvements significantly enhance the scientific rigor and policy relevance of the study. Once again, we thank you for your constructive criticisms, which have been instrumental in improving the manuscript.

**Major Comments:**

Lines 15–80 (Introduction): The manuscript adopts a conventional bottom-up emissions estimation framework without methodological innovation. The research question is descriptive in nature and lacks originality. Similar approaches using MODIS and empirical EF models have been extensively

applied in both national and global contexts.

**Response:** Thank you for your important observation regarding the methodological innovation and scope of our study. We acknowledge that this research employs a classical bottom-up emission estimation framework, which has been widely used in global emission inventories such as GFED and FINN. This choice was made deliberately, as our primary objective is not to propose a new emission modeling algorithm, but rather to construct a long-term (2001-2022), spatially explicit wildfire $CO_2$ emission inventory for China using harmonized and validated local datasets. Our aim is to gain a deeper understanding of wildfire $CO_2$ emissions in China. To clarify these points, we have revised the Introduction (Lines: 74-93 in the revised manuscript) to emphasize the specific ways in which our study adds value to the literature. The details are as follows:

"Traditionally, wildfire emission inventories using population or cropland area weights to allocate total emissions to grid cells have high uncertainties (Streets et al., 2003; Zhang et al., 2013). With the advancement of remote sensing technology, recent studies have shifted to satellite-based estimation methods, using active fire detection or burned area datasets to improve spatial accuracy. Inventories such as GFED (Chen et al., 2023) and the NCAR Fire Inventory (FINN) (Wiedinmyer et al., 2011) rely on satellite-derived fire count data (e.g., active fire product MCD14 ML) or burned area products (e.g., MCD64A1) to infer the timing and location of fire emissions (Giglio et al., 2016, 2018). Although satellite remote sensing has greatly improved the spatial and temporal resolution of fire detection, several practical challenges remain. For example, cloud cover, satellite overpass intervals, fire intensity thresholds, and pixel resolution can result in the underdetection of short-duration or low-intensity fires. To mitigate these limitations, this study integrated multi-source satellite products to enhance the completeness of the fire signal. Additionally, many existing global inventories rely on globally aggregated vegetation data (such as global land cover, and biomass), which further introduces errors, especially in transition zones between cropland and natural vegetation (e.g., forest-agricultural mosaics), where misclassification may lead to overestimation or underestimation of fire emissions.

To overcome these shortcomings, this study integrated China's regionally validated vegetation cover datasets (Xu et al., 2018) multi-source burned area satellite products, and regionally derived biomass data (Hu et al., 2006; Su et al., 2016; Yin et al., 2023) to develop a 500-meter resolution wildfire $CO_2$ emission inventory for China (2001-2022). Additionally, we used spatially weighted regression models to explore the drivers of emission variability and analyzed the impacts of national fire management policies on $CO_2$ emissions. The findings provide insights into the role of governance in shaping fire emissions and offer useful references for future wildfire management strategies. This multi-year emission inventory can also be used in atmospheric transport models to support the development of effective global warming mitigation strategies."

Lines 110–150 (Methods): The emission estimation follows a static multiplicative model (BA × AGB × EF × CF), which does not account for critical factors such as fire behavior, combustion completeness, fuel moisture, or fire weather variability. The combustion efficiency model, based solely on fractional vegetation cover (FVC), is overly simplistic.

**Response:** Thank you for this profound and important comment. We agree that fire behavior is a dynamic and multifaceted process influenced by fuel conditions, combustion stages, and meteorological factors. However, due to the lack of consistent, high-resolution fire dynamics data at the national scale over long time periods, we employed a static bottom-up emission model

(Burned Area × Aboveground Biomass × Emission Factor × Combustion Efficiency), which is consistent with methodologies used in established global inventories such as GFED and FINN. To enhance regional representativeness for China within this framework, we introduced several improvements. As discussed in detail in our response to Comment [Lines 15–80 (Introduction)]. Additionally, in Section 3.5, we conducted a Monte Carlo uncertainty analysis to evaluate the impact of input parameter uncertainties. The results showed that combustion efficiency contributed relatively less to total emission uncertainty compared to burned area and aboveground biomass. We acknowledge that future research could incorporate dynamic fire behavior parameters, such as fuel moisture, fire radiative power (FRP), or weather-driven combustion phases, particularly in regions with dense fire monitoring networks.

Lines 110–330: The manuscript fails to provide uncertainty ranges or confidence intervals for any of the key input variables (BA, AGB, EF, CF). No Monte Carlo simulation, error propagation, or even basic upper/lower bounds are included.

**Response:** Thank you for the reviewer's meticulous attention to the handling of uncertainty. We agree that quantifying and transparently communicating uncertainty is central to emission inventories. A comprehensive Monte Carlo uncertainty analysis has been conducted in the study, and the details are presented in Section 3.5 of the revised manuscript. The details are as follows (Lines: 463-513):

3.5 Uncertainty analysis

[revised manuscript text omitted]

Lines 270–290 (Figure 10): The authors only apply Spearman correlation to assess drivers of emissions, without controlling for multicollinearity or confounding factors. No multivariate regression, GAM, or causal modeling is attempted, leading to misleading interpretations.

**Response:** Thank you for this valuable comment. We fully agree that using only Spearman correlation in the initial version limited the robustness of the analysis. In the revised manuscript, we replaced the correlation-based method with analytical approaches including Ordinary Least Squares (OLS), Geographically Weighted Regression (GWR), and Geographically and Temporally Weighted Regression (GTWR). The details are as follows (Lines: 333-430):

[revised manuscript text omitted]

Lines 95–100 (Data section): MODIS MCD64A1 is known to underestimate small fires, fires occurring under cloudy conditions, and fires occurring on wet ground, all of which are commonly found in different places in eastern and southern China. The authors do not validate this data or compare it to higher-resolution alternatives (e.g., VIIRS, Sentinel-2).

**Response:** Thank you for this important and constructive comment. We fully recognize that the MODIS MCD64A1 burned area product tends to underestimate small, fragmented, or low-intensity fires. To address this known limitation, we corrected the MODIS burned area using a correction factor ($\alpha_i$). The details are as follows (Lines: 123-133):

2.2.1 Burned areas

BA for each vegetation cover type was primarily estimated using the MODIS-MCD64A1 product(Giglio et al., 2018), which provides global monthly burned area estimates. However, it is well established that MODIS-MCD64A1 tended to underestimate small and fragmented fires. To address this issue, we applied scaling factors ($\alpha_i$) to correct the MODIS-derived BA estimates. The scaling factors were derived from the comparison of MODIS-derived BA with two independent global burned area datasets: the FireCCI51 product (Lizundia-Loiola et al., 2020) released by ESA (http://cci.esa.int/data) and a global GFED500 product (Van Wees et al., 2022), with their specific values provided in Table S1. The corrected burned area for each land cover type was obtained by multiplying the MODIS-derived BA values by the corresponding scaling factor. This correction accounts for the known systematic underestimation of small and fragmented fires by the MODIS MCD64A1 product.

$$BA_{\text{corrected},i} = BA_{\text{MODIS},i} \times \alpha_i \tag{2}$$

where subscript $i$ denotes vegetation type.

In the revised manuscript, we also analyzed the uncertainty in BA. We compared the BA derived from MODIS in 2015 with multiple independent datasets, including FireCCI51 (250 m resolution), GFED (500 m resolution), GABAM (30 m resolution), and the FINN dataset. Detailed discussions are presented in Section 3.5 "Uncertainty Analysis" of the revised manuscript, and We have also provided the detailed content of Section 3.5 "Uncertainty Analysis" (Lines: 463-513) in the comments [Lines 110–330].

Lines 155–265 (Figures 2–9): Results are largely descriptive summaries of emission patterns over time and space. Figures are repetitive and lack analytical depth. There is little attempt to explain spatial heterogeneity beyond surface-level summaries.

**Response:** Thank you for your valuable suggestion. In response, we have revised the Results and Discussion sections to provide a more in-depth interpretation of the spatiotemporal emission patterns. We further explain the regional differences in the peak months of cropland fire emissions. The details are as follows (Lines: 263-269):

"In contrast, the spatial distribution of emissions from cropland fires varied significantly across different months and was likely influenced by policy management (Fig.4). High emissions in March and April were concentrated in NE region, while emissions in May and June were primarily associated with the NC region. The regional difference in peak emission months can be attributed to distinct cropping systems and climatic conditions. In the NE region (e.g., Heilongjiang and Jilin), cold winters and delayed spring thaw often push straw burning activities into March-April, following the autumn harvest. In contrast, the NC region (e.g., Anhui, Henan, Jiangsu) practices a double-cropping system of winter wheat and summer maize, where wheat is harvested in May–June, and burning of straw residues is typically observed during this transition period."

To address spatial heterogeneity, we applied OLS, GWR, and GTWR regression models (Section 3.4) to quantify the spatially varying impacts of climatic variables (temperature, precipitation, humidity, sunshine duration) and socioeconomic factors (GDP, population density) on emissions. We have also provided the detailed content of Section 3.4 "The impact of factors on wildfires in China" (Lines: 333-430) in the comments [Lines 270–290 (Figure 10)].

We have also reduced potential redundancy by reorganizing and optimizing the figures and tables. We hope these additions and improvements address the reviewer's concerns regarding analytical depth and help clarify the mechanisms driving the spatiotemporal patterns of wildfire $CO_2$ emissions.

Lines 295–310 (Implications): The manuscript claims that policies reduced emissions (e.g., crop burning bans) but offers no empirical evaluation, no event-based analysis, and no causal testing (e.g., Difference-in-Differences or time series break analysis).

**Response:** Thank you for your valuable suggestion. We acknowledge that the manuscript currently does not employ formal causal inference methods such as difference-in-differences (DID) or interrupted time series analysis. While these methods are valuable, their application in this study is constrained by several factors. Policy interventions such as open-burning bans or forest fire prevention plans are not implemented instantaneously or uniformly; they typically undergo a gradual process of adoption, enforcement, and local adaptation that varies by region and administrative level. This lagged and heterogeneous implementation pattern makes it difficult to apply event-based causal attribution designs with clear cutoff points. In the revised manuscript, we have linked the observed emission trends to major regulatory milestones (e.g., the 2015 revision of the Air Pollution Prevention and Control Law, and the 2009 and 2016 National Forest Fire

Prevention Plans) (Figure 12). Although these associations do not constitute strict statistical causality, they provide plausible policy-related explanations. In the newly revised manuscript, we have removed the original Section 3.5 "Implications" and incorporated the revised content into Section 3.4 (Lines: 431-462). The details are as follows:

"Although the GTWR model indicated that climatic and socioeconomic variables such as TMP, RH, and GDP explained the spatial variation in wildfire $CO_2$ emissions, the overall model performance remains moderate for forest and shrub fires, with particularly low explanatory ability for grassland fires ($R^2 = 0.31$). This gap suggests that, beyond natural and socioeconomic factors, other key drivers may have been omitted. Multiple studies (Gao et al., 2023; Kelly et al., 2013; Phillips et al., 2022; Xie et al., 2020) highlight the substantial impact of fire management policies on $CO_2$ emissions. For instance, Phillips et al. (2022) showed that the marginal abatement cost of avoiding fire-related $CO_2$ emissions through fire management is comparable to or even lower than that of many other climate mitigation strategies.

In China, the role of policy is particularly significant. Wu et al. (2018) reviewed 51 crop straw management regulations issued between 1965 and 2015, with 34 implemented after 2008. The timing of these intensive regulatory efforts closely aligns with key turning points in emission trends (Fig. 12). For cropland fires, annual $CO_2$ emissions increased from 8.2 Tg/year during 2001-2005 to 26.2 Tg/year during 2010-2016, but began to decline following the revision of the Air Pollution Prevention and Control Law in 2015 and the launch of the Air Pollution Action Plan in 2013. Similarly, after the implementation of the National Forest Fire Prevention Plans in 2009 and 2016, $CO_2$ emissions from forest, shrub and grassland fires dropped from 38.1 Tg/year (2006–2009) to 13.3 Tg/year (2017–2022). Jin et al. (2022) further estimated that over 80% of wildfire-related $CO_2$ emissions could be avoided under effective fire management. These findings strongly indicate that policy management plays a critical role in wildfire $CO_2$ emissions.

Notably, northeastern China is the only region where cropland burning has increased in recent years, highlighting the need for adaptive rather than restrictive policies. As one of China's major grain-producing regions, Northeast China generates large volumes of straw. Harsh winters and short windows for straw return or removal, combined with long-established farming practices, have made complete bans on straw burning particularly challenging. Prior to strict open-burning prohibitions, farmers often burned straw in a dispersed, low-intensity manner, making detection by satellite-based fire products difficult, potentially resulting in systematic underestimation of early emissions. After the implementation of strict bans, facing growing pressure from unprocessed straw accumulation, therefore, some local governments adopted more adaptive fire management policies, such as designating burning windows under favourable meteorological conditions. These "limited and concentrated burning periods" led to spatiotemporally clustered fire events that were more easily captured by remote sensing. In recent years, the Chinese government has also promoted the scientific incorporation of straw into soils, off-field collection, and the industrial utilization of crop residues in Northeast China. These efforts highlight the significant role of policy in shaping emission trends from agricultural burning, particularly in regions where environmental constraints and traditional farming practices pose unique challenges."

[Figure]

Figure 12: Temporal trends in annual $CO_2$ emissions from cropland burning and forest, shrub, and grassland fires in China (2001–2022), and key national policy milestones related to fire and air pollution control.

**Minor Comments:**

L120-L125 (Methods): The AGB data for forests/shrubs are sourced from Su et al. (2016) and Yan et al. (2023), which likely differ in methodology and spatial resolution. No effort is made to harmonize these datasets, risking inconsistencies in long-term trends.

**Response:** Thank you for this important comment. We agree that the consistency of input datasets is crucial for long-term trend analysis. In this study, we used the forest aboveground biomass (AGB) dataset from Su et al. (2016) for the period 2001–2012 and the dataset from Yin et al. (2023) for 2013–2022. These datasets were selected to reflect annual changes in aboveground biomass rather than assuming constant AGB throughout the study period. To assess the compatibility of these two datasets, we compared their annual average values and found that the AGB from Su et al. (2016) averaged approximately 100 t/ha, while that from Yin et al. (2023) averaged 107 t/ha—a difference of about 7%, which falls within the uncertainty range (±50%) for forest biomass estimates noted by Yin et al. (2023). Therefore, we consider the transition between these two datasets acceptable and unlikely to distort long-term emission trends.

Regarding shrub, further analysis revealed that directly using the forest AGB dataset in the previous manuscript might not accurately reflect the biomass characteristics of Chinese shrubs. To improve the regional applicability of our estimates, we replaced the previous data with locally derived shrub biomass density values based on Hu et al. (2006), as detailed in Table S2 of the revised manuscript. This substitution ensures more accurate and representative AGB inputs for shrub fire emission

estimation in China. The uncertainty analysis regarding AGB is provided in Section 3.5 "Uncertainty Analysis". We have also provided the detailed content of Section 3.5 "Uncertainty Analysis" (Lines: 463-513) in the comments [Lines 110–330].

L120-L125 (Methods): The NDVI-based exponential model (Gao et al., 2012) may saturate in high-biomass regions, leading to underestimation. The authors do not address this limitation or compare their AGB estimates with field measurements or independent datasets (e.g., LiDAR), casting doubt on the reliability of the method.

**Response:** We appreciate the reviewer's thoughtful comment. We acknowledge that NDVI-based exponential models can exhibit saturation effects in high-biomass regions, potentially underestimating aboveground biomass (AGB), especially in dense grassland ecosystems. To assess this concern, we conducted additional analysis using the corrected model proposed by Hu et al. (2024), which specifically addresses NDVI saturation in alpine meadow grasslands in China. This model introduces a revised NDVI-AGB function to mitigate underestimation under high vegetation density conditions.

$$N DVI\text{adj} = \begin{cases} \text{NDVI}, (0 \leqslant \text{NDVI} < 0.728831) \\ 0.0428 \times \dfrac{1 + \text{NDVI}}{1 - \text{NDVI}} + 0.4552, (0.728831 \leqslant \text{NDVI} \leqslant 1) \end{cases}$$

$$y = 5908.5x - 2198.9 \left( \frac{kg}{hm^2} \right)$$

We recalculated AGB using Hu's modified model and compared it to the original Gao et al. (2012) model. The results showed a minimal difference in average AGB estimates: 210 g/m$^2$ from Gao's model vs. 214 g/m$^2$ from Hu's model. The difference falls well within the reported uncertainty bounds of both models (±62.5 g/m$^2$ in Gao et al., ±85 g/m$^2$ in Hu et al.). Moreover, Gao et al.'s model was based on field sampling across a wider range of vegetation types, including arid grasslands, meadow steppes, wetlands, and sandy vegetation, which better matches the diverse ecosystems included in our national-scale analysis. Given the small difference in results and the broader applicability of Gao's model across multiple vegetation types, we have retained Gao et al. (2012) as our baseline model. Additionally, uncertainties associated with AGB estimation have been explicitly assessed through Monte Carlo simulation in Section 3.5. We have also provided the detailed content of Section 3.5 "Uncertainty Analysis" (Lines: 463-513) in the comments [Lines 110–330].

L135-L40 (Methods): The combustion efficiency (CF) equations rely on FVC thresholds derived from Hély et al. (2003), which were developed for African savannas. The authors do not validate whether these thresholds are appropriate for Chinese ecosystems (e.g., boreal forests in Heilongjiang vs. arid grasslands in Inner Mongolia and tropical forests in Hainan and Yunnan). This raises concerns about model transferability and regional accuracy.

**Response:** Thank you for the reviewer's insightful observation. We fully recognize that the empirical relationship between combustion efficiency (CE) and fractional vegetation cover (FVC) proposed by Hély et al. (2003) was originally derived from African savanna ecosystems. In this study, we applied this approach to forests and grasslands primarily to enable spatially estimation of CE based on vegetation structure, which is particularly useful for long-term national-scale emission inventories. Although we acknowledge the limitations of directly applying the Hély model to all Chinese ecosystems. For shrubs and croplands, we did not use the Hély equation but instead adopted fixed CE values based on China-specific and global literature (e.g., Junpen et al., 2020; Zhou et al.,

2017; He et al., 2015).Additionally, we conducted a Monte Carlo uncertainty analysis (Section 3.5) to quantify the impact of CE variability on total emission estimates. The results show that while CE introduces some uncertainty, its relative contribution to total emission uncertainty is lower compared to factors such as aboveground biomass (AGB) and burned area (BA). uncertainties associated with CE estimation have been explicitly assessed through Monte Carlo simulation in Section 3.5. We have also provided the detailed content of Section 3.5 "Uncertainty Analysis" (Lines: 463-513) in the comments [Lines 110–330].

L220-225 The term "high-confidence hotspot" is used without defining confidence thresholds (e.g., 95% confidence). Clarify how hotspots were statistically determined to be of high-confidence.

**Response:** Thank you for pointing this out. In the revised manuscript, we have clarified the statistical criteria used to define "high-confidence hotspots". We have updated the relevant text in Section 2.3.2 "Hot spot analysis" (Lines: 199-214). The details are as follows:

2.3.2 Hot spot analysis

While Moran's I provides a global measure of spatial autocorrelation, it does not explicitly identify localized clusters of high or low values. To address this limitation, the Getis-Ord Gi* statistic (Getis and Ord, 1992) is commonly used to identify statistically significant hotspots and coldspots within spatial datasets. Unlike Moran's I, which captures both positive and negative spatial autocorrelation, the Gi* statistic focuses on detecting concentration patterns of high or low values within the study area. The Getis-Ord Gi* statistic is defined as:

$$G_i^* = \frac{\sum_j w_{ij} x_j - \bar{x} \sum_j w_{ij}}{S \sqrt{\left[\frac{n \sum_j w_{ij}^2 - \left(\sum_j w_{ij}\right)^2}{n-1}\right]}} \tag{7}$$

Where $G_i^*$ is the Getis-Ord Gi* statistic for location $i$; $x_j$ is the observed value at location $j$ (e.g., $CO_2$ emissions); $\bar{x}$ is the global mean of the observed variable; $w_{ij}$ is the spatial weight matrix, representing the spatial relationship between locations $i$ and $j$; $n$ is the total number of spatial units; $S$ is the standard deviation of the observed values.

The Gi* statistic is essentially a ratio that compares the local sum of a variable within a specified distance to the global sum, adjusted for the number of spatial units and their spatial relationships. High positive Gi* values indicate clusters of high values (hotspots), while low negative Gi* values indicate clusters of low values (coldspots). Locations with Gi* values near zero indicate random spatial patterns without significant clustering. Statistical significance is assessed using Z-scores and corresponding $p$-values. In this study, Gi* analysis was used to detect persistent high- and low-emission clusters of wildfire $CO_2$ emissions across China from 2001 to 2022. The results provided spatially explicit insights into emission patterns and their temporal dynamics.

L235-240 "Human activities and fire management may affect cropland fire emissions more significantly, resulting in more significant variability..." Replace the second "significant" with "pronounced" for clarity.

**Response:** Thank you for your suggestion. We appreciate the comment regarding word choice; however, the sentence in question (Lines: 235–240 in the original manuscript) has been removed during the revision process as part of a broader restructuring of the analysis.

L205 (Table 1): While Moran's I confirms clustering, the authors do not explore why spatial clusters exist (e.g., links to regional policies, economic activities, climate zones, fire spread, etc.). This limits the utility of spatial analysis for informing targeted mitigation strategies.

**Response:** Thank you for your valuable suggestions. In the revised manuscript, we have expanded the analysis and interpretation of the spatial clustering patterns of wildfire $CO_2$ emissions. The revised content can be found in Section 3.3 "Spatiotemporal Variations in $CO_2$ Emissions" (Lines: 277-332). The details are as follows:

[revised manuscript text omitted]

L285-290 (Table 2): Citations for "Chen et al. (2022)" (GDP data) and "LandScan" (population density) are missing from the reference list. Ensure all data sources are fully cited.

**Response:** Thank you for pointing this out. We have now added complete citations for both the GDP dataset ("Chen et al., 2022") (Line: 603) and the LandScan Global Population Dataset in the revised reference list (Line:597).

L290 (Figure 10): Spearman's correlations assume independence of observations, but spatial autocorrelation violates this assumption. The reported significance levels (p-values) may be inflated, leading to spurious correlations. Spatial regression or geographically weighted regression (GWR) should be used instead.

**Response:** Thank you for this important and technically valid observation. To address this, we have revised the analytical framework, replacing the bivariate correlation approach with spatial regression models, and conducted a comparative analysis of three models including Ordinary Least Squares (OLS), Geographically Weighted Regression (GWR), and Geographically and Temporally Weighted Regression (GTWR). We have removed the original Spearman correlation results and replaced Figures 8-11 with updated spatial regression outputs and interpretation. This improvement has significantly enhanced the robustness and interpretability of our findings. These additions are now discussed in Section 3.4 of the revised manuscript. We have also provided the detailed content of Section 3.4 "The impact of factors on wildfires in China" in the comments [Lines 110–330].

Lines 315–330 (Conclusion): The conclusion restates earlier findings without elevating the discussion. Consider framing in terms of implications for global carbon inventories, LULUCF reporting, or wildfire mitigation strategies.

**Response:** Thank you for your insightful suggestions. In the revised manuscript, we have carefully revised and expanded the conclusion section to enhance its interpretive depth and policy relevance. The details are as follows (Lines: 539-565):

4 Conclusion

This study developed a comprehensive inventory of wildfire $CO_2$ emissions across China from 2001 to 2022, capturing significant spatiotemporal variations among different vegetation types. Results showed that cropland and forest fires were the primary contributors to national wildfire emissions. Forest and shrub fire $CO_2$ emissions exhibited a declining trend, grassland fire $CO_2$ emissions remained relatively stable, and cropland fire $CO_2$ emissions showed an increasing trend. GTWR analysis revealed that shrub fire $CO_2$ emissions exhibited the highest predictive performance

($R^2 = 0.87$), with climatic factors (particularly temperature and humidity) being the main influencing factors, and limited temporal variation. In contrast, forest and cropland fire $CO_2$ emissions were significantly influenced by the spatiotemporal heterogeneity of both climatic and socioeconomic factors. Grassland fire $CO_2$ emissions exhibited the lowest model explanatory power ($R^2 = 0.31$), suggesting that their emissions may largely depend on drivers not included in the current model.

Our findings underscore the critical role of policy interventions in shaping wildfire emissions in China. The observed declines in most regions aligned with the implementation of national fire control and air pollution reduction programs. However, northeastern China remained an exception, with cropland fire $CO_2$ emissions continuing to increase in recent years. This trend highlighted the limitations of blanket burning bans and the necessity of adaptive fire management. Although forest fire $CO_2$ emissions had been reduced through strengthened fire prevention measures, northeastern China remained vulnerable to extreme fire events triggered by drought or lightning. Shrub fire $CO_2$ emissions, primarily driven by climatic factors, underscore the importance of strengthening early-warning systems.

Although wildfire emissions are classified as "natural disturbances" under IPCC guidelines for LULUCF and are often excluded from national emission inventories, the results demonstrated that these emissions were substantial and closely tied to policy and land management practices. The pronounced interannual variability and spatial heterogeneity suggested that future climate extremes, land-use changes, or fire policy adjustments could significantly alter regional carbon dynamics.

Compared with global emission inventories (GFED, FINN, QFED, GFAS), the estimates in this study were generally lower. Although remote sensing data might underestimate some cropland fires, this study characterized wildfire $CO_2$ emission patterns in China by integrating multi-source burned area products, localized biomass data, and high-resolution land cover classifications. Future research should further refine burned area identification, optimize parameters such as emission factors and combustion efficiency, bridge observational gaps, and incorporate transboundary fire dynamics to ensure more comprehensive and accurate regional emission accounting.

Lines 360 (References): Several references (e.g., Cao et al., 2004) lack English translations or full journal details. Ensure uniform citation style. Furthermore, while some people may be able to read the article, I am not sure if all of the reviewers can understand it fully.

**Response:** Thank you for this helpful suggestion. In the revised manuscript, we have carefully reviewed and updated all Chinese-language references to include English translations of titles, complete journal and publication details, and a clear indication that the source is "in Chinese". We have followed the citation style commonly recommended by the journal for non-English references.